evolution, genetics, genomics

microevolution, genomic prediction, cryptic evolution, genomic estimated breeding value, soay sheep

**Author for correspondence:**
J. Slate
e-mail: j.slate@sheffield.ac.uk

One paper of a special feature 'Wild Quantitative Genomics: the genomic basis of fitness variation in natural populations' edited by Susan Johnston, Nancy Chen and Emily Josephs.

# Using genomic prediction to detect microevolutionary change of a quantitative trait

D. C. Hunter[1,3], B. Ashraf[1,4], C. Bérénos[2], P. A. Ellis[2], S. E. Johnston[2], A. J. Wilson[5], J. G. Pilkington[2], J. M. Pemberton[2] and J. Slate[1]

[1]School of Biosciences, University of Sheffield, Sheffield S10 2TN, UK
[2]Institute of Evolutionary Biology, University of Edinburgh, Edinburgh EH9 3FL, UK
[3]School of Biology, University of St Andrews, St Andrews KY16 9ST, UK
[4]Department of Anthropology, Durham University, Durham DH1 3LE, UK
[5]Centre of Ecology and Conservation, College of Life and Environmental Sciences, University of Exeter, Penryn TR10 9FE, UK

JS, 0000-0003-3356-5123

Detecting microevolutionary responses to natural selection by observing temporal changes in individual breeding values is challenging. The collection of suitable datasets can take many years and disentangling the contributions of the environment and genetics to phenotypic change is not trivial. Furthermore, pedigree-based methods of obtaining individual breeding values have known biases. Here, we apply a genomic prediction approach to estimate breeding values of adult weight in a 35-year dataset of Soay sheep (*Ovis aries*). Comparisons are made with a traditional pedigree-based approach. During the study period, adult body weight decreased, but the underlying genetic component of body weight increased, at a rate that is unlikely to be attributable to genetic drift. Thus cryptic microevolution of greater adult body weight has probably occurred. Genomic and pedigree-based approaches gave largely consistent results. Thus, using genomic prediction to study microevolution in wild populations can remove the requirement for pedigree data, potentially opening up new study systems for similar research.

## 1. Introduction

When directional selection on a single trait is carried out in an experimental evolution study, or in animal and plant breeding, the response to selection is predictable. The breeder's equation [1], $R = h^2S$, predicts the response ($R$) as the product of the strength of selection ($S$) and the narrow-sense heritability ($h^2$), and it usually gives a reasonably accurate estimate of the actual observed response [2]. However, in more complex systems, such as in wild populations, the response is much harder to predict [3,4]. The reasons why a trait may not evolve as expected include: (1) unmeasured genetic correlations between the trait and other fitness-related traits [3,5]; fluctuating environmental conditions covarying with (2) the heritability of the trait [6] or (3) the strength of selection [7] (or both [8]); and (4) cryptic microevolution [9], where the trait actually *has* responded to selection, but a change in environmental conditions has caused the phenotypic trend to mask the underlying genetic trend. More generally, the term 'paradox of stasis' has been used to describe the absence of expected response to selection [4]. The upshot is that identifying microevolutionary responses to natural selection is non-trivial.

One way to detect microevolution in a population is to estimate each individual's 'genetic merit' or breeding value, and test whether population-wide breeding values have changed temporally, in line with predictions from measured

selection differentials. If phenotypic and multigenerational pedigree data are available then estimated breeding values (EBVs) can be obtained using restricted maximum-likelihood (REML) procedures in linear mixed models—'animal models' [10–12]. Exploring temporal trends in EBVs rapidly gained popularity once animal models began to be applied to studies of natural populations [9,13,14]. Unfortunately, it was not long before problems and biases with the approach were identified [15,16]. First, pedigrees in natural populations are usually small and unbalanced, relative to those used in applied breeding. Therefore, many individuals lack 'connectedness'—they have relatively few phenotyped relatives in the dataset. EBVs of these individuals are heavily dependent on the individual's phenotype, and so environmental influences on the phenotype can be confounded with genetic effects, causing error and bias in the EBV [16]. In studies looking at microevolutionary trends, this can cause spurious evidence for microevolutionary change, especially when 'unconnected' individuals are clustered at the beginning or end of the time series and have experienced similar environments. A second major problem is that EBVs from animal models have sometimes been treated as point estimates, with any uncertainty or error in their estimate being ignored. Ignoring this uncertainty causes temporal trends in EBVs to be anti-conservative (i.e. prone to false positive inferences), especially as errors tend to be correlated among relatives [17].

To overcome the problems described above, several best practice steps have been proposed. First, fitting terms such as year of birth or measurement in an animal model (de-trending) should reduce the risk of temporal environmental heterogeneity causing spurious trends in EBVs, although it may not eliminate it completely [16]. Second, taking a Bayesian approach to estimate breeding values allows the uncertainty in estimates to be accommodated in the analysis of temporal trends [17]. Bayesian animal models return a posterior distribution of EBVs, rather than a point estimate, and by regressing every posterior sample of cohort mean EBVs on year, the slope can be estimated with an appropriate, confidence interval [17]. In their paper demonstrating this approach, Hadfield and colleagues [17] pointed out that there are two different questions that can be asked with the posterior distributions of EBVs. Is the microevolutionary trend statistically significant? And is the trend of greater magnitude than can be expected due to genetic drift? The second question is biologically more relevant, in the context of genetic response to selection, and can be tested by simulating expected changes in breeding values due to drift alone. Usually, this is done by sampling from the posterior distribution of the trait's additive genetic variance and randomly assigning the founders in the pedigree a simulated breeding value. Simulated breeding values are then transmitted down the pedigree, by assigning offspring a midparental EBV. The simulated EBVs can be regressed against time to estimate the microevolutionary trend. If the process is repeated for each posterior estimate of the additive genetic variance, then a null distribution of temporal trends under a scenario of no selection (and thus no response) is generated (i.e. the expectation due to genetic drift).

In their paper highlighting how microevolutionary trends derived from point estimates of breeding values tended to be anticonservative, Hadfield and colleagues [17] illustrated the problem with two empirical datasets. One of these was an example of apparent cryptic microevolution of adult body size in a free-living population of feral Soay sheep [18]. In the study population, adult weight is positively associated with increased winter survival and is heritable [19–21], and therefore should be increasing in response to directional selection. In fact, during the period studied by Wilson and colleagues (1985–2005), adult body weight declined over time by around 100–200 g per year [18]. By contrast, during the same period, animal model-derived EBVs for adult weight increased by around 5–10 g per year. In other words, microevolution was proceeding as expected but was masked by non-genetic effects causing body sizes to be smaller [20]. The contrast between phenotypic and genetic trends is explained by smaller animals having a better survival rate in the more recent years of the study, perhaps because harsh winters have become less common and grass growth rates have improved [20]. Thus, smaller animals that once would have died have a better chance of surviving and reducing the population mean weight. Birth year and capture year were fitted in the animal models, thus de-trending the data for possible environmental effects on different cohorts, prior to estimating the temporal trends. Furthermore, the phenotypic and genetic trends were in opposite directions, which is generally considered to be a sign that apparent microevolution is not being driven by an unappreciated contribution of an environmental effect on phenotype [16]. However, the original analysis was anti-conservative as inference on the genetic trend failed to properly account for uncertainty in EBV estimation. Re-analysis by Hadfield et al. using a Bayesian method to incorporate the posterior distributions of EBVs yielded an estimate of the temporal increase in breeding values that was quantitatively unchanged, but no longer statistically significant. Thus, the more conservative (and correct) analytical approach [17] no longer supported an evolutionary response to selection on larger adult body size.

In this paper, we revisit the question of whether adult weight in Soay sheep is responding to natural selection, but with several potential improvements over previous investigations. First, we use a larger dataset (electronic supplementary material, table S1) that extends the time series studied by Wilson by a further 10 years. Second, and perhaps most obviously different to previous work, we use genomic prediction rather than a pedigree to estimate breeding values. Genomic prediction [22], is a tool widely used in animal and plant breeding to estimate quantitative genetic parameters from marker data. More recently, genomic prediction has started to become adopted by researchers working on wild populations [23–26]. It works by exploiting the fact that when marker density is sufficiently high, some typed SNP markers are in linkage disequilibrium (LD) with unknown causal loci, and the contribution of each SNP (and the unknown loci it tags) to phenotypic variation can be estimated. By estimating SNP effects in one part of the dataset (the training population) and using those estimates to predict breeding values in a second dataset of genotyped but not phenotyped individuals (the test population) it is possible to obtain genomic estimated breeding values (GEBVs) that are independent of the focal individual's phenotype. Furthermore, GEBVs are less influenced by the number of phenotyped close relatives a focal individual has in the dataset, compared to animal model-derived EBVs [27]. Genomic prediction models can include known fixed and random effects such as sex, age, year of birth and year of measurement. We use a Bayesian method to estimate the GEBVs, so posterior distributions of GEBVs can be used to account for uncertainty in their estimates, in the same

way, that Hadfield recommends for EBVs derived from Bayesian animal models of pedigree data. We compare our results from genomic prediction-derived EBVs with those obtained from the traditional pedigree-based animal model approach. Third, we use a 'gene-dropping' simulation approach, to formally test whether any temporal patterns are likely to be due to a response to selection or are explainable by genetic drift. However, because the genomic prediction method estimates the contribution of each SNP to variation in weight, the genetic drift simulations are explicitly based on a description of the trait's genetic architecture (the number and effect size of causal loci) rather than assuming an infinitesimal model. Here, haplotypes of causal loci are assigned to founders, and offspring inherit alleles at each locus according to Mendelian inheritance and empirically estimated recombination rates between linked loci. Thus, these simulations are analogous to methods that simulate the inheritance of breeding values down a pedigree.

Our intention is that, in addition to revisiting a well-known, but still unresolved, case study of possible cryptic microevolution, this paper will illustrate a genomic prediction-based framework to study microevolutionary change that can be readily adopted by other researchers working on evolution and adaptation in natural populations.

## 2. Methods

### (a) The study population
The Soay sheep is a primitive feral breed inhabiting the St Kilda archipelago, off the northwest coast of Scotland. Since 1985, the population in the Village Bay area of the largest island, Hirta (57 °48′ N, 8 °37′ W), has been the subject of a long-term individual-based study [28]. Most of the sheep residents in the study area are ear-tagged and weighed shortly after birth and followed throughout their lifetime. Ear punches and blood samples suitable for DNA analysis are collected at tagging. During the annual 'catch' in August, sheep are captured and morphological measurements are taken. Winter mortality is monitored, with the peak of mortality occurring at the end of winter/early spring, and approximately 80% of all deceased sheep are found. To date, extensive life-history data have been collected for over 10 000 sheep. More details of the study can be found elsewhere [28].

### (b) Phenotypes
Weight measurements were taken on live animals, as described elsewhere [21,29]. Weight data were restricted to animals captured in August, that were at least 28 months old, to remove most complications of growth [21,29]. Weight data were pre-adjusted for non-genetic factors for the genomic prediction analyses (see below). The inclusion of measurements taken outside of August would not have resulted in a much larger sample size, but would have complicated analyses as animals lose weight at different rates later in the year. Note that Wilson *et al.* [18] included animals aged 0 months, 4 months and 16 months in their analyses [18], but weight has lower additive genetic variance and heritability at these ages. Given that we were using repeated measurements taken across an individual's lifetime, rather than running age-specific models, it was deemed prudent to restrict the analyses to older animals. August weights at different adult ages have genetic correlations of approximately 0.99 [18].

### (c) Genotyping
Genotyping of the population was performed using the Illumina Ovine SNP50 beadchip array, developed by the International Sheep Genomics Consortium (ISGC) [30] Genotyping was performed at the Wellcome Trust Clinical Research Facility Genetics Core (Edinburgh, UK). Details about the genotype calling and quality control of the data and links to the genotype data are available elsewhere [21,31]. Briefly, pruning of SNPs and individuals was performed using Plink v. 1.9 [32]. Only autosomal SNPs were analysed, as BayesR cannot distinguish between autosomal and sex-linked loci (the sheep X chromosome represents approximately 5% of the total genome). SNPs were removed if they had more than 2% missing data, a minor allele frequency less than 0.01 or a Hardy-Weinberg Equilibrium Test $p$ value < 0.00001. Individuals were retained if they were typed for at least 95% of SNPs. After pruning, there were 35 882 SNPs in the dataset. 1168 individuals were genotyped and phenotyped, and there were an additional 5627 animals that were genotyped but not phenotyped (because they died or emigrated from the study area before the age of 28 months), whose weight GEBVs could nonetheless be estimated as a test population (see below).

### (d) Genomic prediction of weight GEBVs
Soay sheep adult weight is a polygenic trait with a moderate heritability and no individual loci were significant in previous GWAS [21,26,29]. Genomic prediction of GEBVs was performed using the BayesR method [33] implemented in the BayesR v. 0.75 software package [34]. We have previously shown that BayesR-derived GEBVs of Soay sheep morphological traits, including adult weight, have a high accuracy (approx. 0.64). BayesR models SNP effects as a mixture of distributions of different effect sizes, including one of zero effect. We used the default settings that model four distributions of effect size of 0, 0.0001, 0.001 and 0.01 of the phenotypic variance. Dirichlet priors for the number of pseudo-observations (SNPs) in each distribution were set to 1, 1, 1 and 5. Priors for the genetic and residual variances were chosen as a scaled inverse-chi squared distribution with scaling parameters of 1.2 and 2.5 and degrees of freedom set to 10. Our previous work has shown that GEBVs are not sensitive to the model parameters used [26]. The MCMC chain was run for a total of 120 000 iterations with a burnin of 20 000 and a thinning interval of 10, meaning there were 1000 posterior samples of the GEBVs for each individual.

In the BayesR models all phenotyped and genotyped animals ($n = 1168$) were treated as a training population and all animals for whom we had genotypes but no phenotypes ($n = 5627$) were the test population. The phenotypes used in the BayesR analysis were obtained by first fitting a linear mixed model (see [26]) that included individual identity (to account for repeated measures), birth year and capture year as random effects, and sex and age as fixed effects. The random effect of individual identity was used as the phenotype.

### (e) Pedigree-based prediction of weight EBVs
Previous analyses of Soay sheep microevolutionary trends for weight have used EBVs derived from animal models of pedigree and phenotype data [17,18]. While the main motivation of this study was to explore the potential of GEBVs to study microevolution, meaningful comparison with earlier studies requires a consistent approach, and so we also estimated EBVs with the pedigree data. Pedigree-derived EBVs were obtained using the R package MCMCglmm v. 2.32 [15]. The model included sex and capture age as fixed effects, and birth year and capture year as random effects (electronic supplementary material, table S2). Permanent environment effects were modelled by fitting identity as a random effect and the additive genetic variance was estimated by fitting a relationship matrix derived from the pedigree. The model was run for 600 000 iterations with a burn-in of 100 000 iterations. The posterior distribution of parameter estimates was obtained by sampling every 500th

iteration after the burn in. Note that the MCMCglmm runs required more iterations, a longer burn-in and less frequent sampling than the BayesR analyses because more terms were estimated in the MCMCglmm models. However, both types of analysis finished with 1000 posterior samples. Subsequent analyses of microevolutionary trends were performed only on animals who were genotyped, to ensure complete consistency with the analyses that used GEBVs.

## (f) Measuring microevolutionary change

To test for microevolutionary change we explored whether EBVs for weight changed as a function of birth year. The cohort mean EBVs were regressed against birth year, weighting each year by the sample size. We used all 1000 posterior samples from the BayesR (genomic EBVs)/MCMCglmm (pedigree EBVs) chain and determined a 95% confidence interval for the slope of cohort mean EBV on year (i.e. the approach advocated by Hadfield *et al.* [17] and subsequently adopted elsewhere [35–37]). The probability of stasis or a decline in EBVs was determined as the proportion of the 1000 models where the slope was less than or equal to 0. We also explored an alternative approach of using individual EBVs and fitting linear mixed models with birth year included as both a fixed effect and as a random effect. The birth year random term was fitted to account for between-year heterogeneity of variances in EBVs. In fact, the birth year random effect explained less than 1% of the variance in GEBVs and about 4% of the variance in pedigree EBVs, and the mixed model approach yielded almost identical results to the regressions of cohort mean EBVs on year (electronic supplementary material, tables S4, S5).

The main results section reports findings from both the training and test populations combined, in all cohorts from 1990 onwards. We omitted earlier years because they had less than 100 individuals (electronic supplementary material, table S1) and earlier cohorts were treated slightly different in the gene-dropping simulations of drift (see below). However, to enable a comparison with the previous work, we also report trends from the same years studied by Wilson (1985–2005). In the electronic supplementary material, we show results from 1980 onwards and also from 2005 onwards (i.e. in the cohorts born since Wilson's study [18]). Similarly, we compare results from models that contained training and test population individuals with those that just contained the test population. The main findings are not sensitive to the choice of cohorts or populations used (electronic supplementary material, tables S4 and S5).

## (g) Simulating microevolutionary change under genetic drift

Simulations to explore microevolutionary changes expected under genetic drift used an approach where breeding values were simulated in the pedigreed population, similar in concept to those introduced by Hadfield *et al.* [17] and adopted by others [37]. However, whereas previous methods have transmitted breeding values from parents to offspring down a pedigree by assigning a midparental breeding value to offspring, we dropped individual SNPs down the pedigree and then calculated GEBVs from the estimated effect sizes of SNPs. This has the advantage that instead of assuming a near-infinitesimal polygenic architecture, the GEBVs are predicted using empirical estimates of the number and effect size of causal loci, as well as realistic recombination fractions between them. Gene-dropping was performed using the SimPed program [38]. SimPed can handle SNP genotypes or haplotype blocks of linked SNPs. To ensure that realistic levels of LD between linked SNPs was present in the simulated datasets, we used

haplotypes, after first phasing the real data. Phasing was performed and missing genotypes were imputed using Beagle v. 5.0 [39], assuming an effective population size of 200 [30]. Phased haplotypes were then randomly assigned to founder individuals in the Soay sheep pedigree. Individuals born before 1990 were treated as founders. The SimPed gene-dropping simulations use known linkage distances between all of the SNPs to ensure that realistic amounts of recombination occur during each meiosis. We used previous estimates of recombination fractions in the Soay sheep pedigree [31]. Because SimPed can only run one chromosome of markers at a time, we concatenated all of the SNPs from different chromosomes to create one 'super chromosome', but assumed a recombination fraction of 0.5 between the last SNP on one chromosome and the first SNP on the next chromosome. This effectively ensures independent segregation of unlinked chromosomes. In cases where an individual was missing parental data, a haplotype was randomly assigned from an individual of the correct sex that was born between 2 and 10 years prior. Handling missing data in this way ensures that complete genotypes were simulated for all individuals while accounting for any temporal changes in allele frequencies in the population. Alleles are passed from parents to offspring following Mendelian rules of independent assortment and recombination. Each SimPed run generates genotypes at every SNP in every individual in the pedigree. Multilocus genotypes were converted to GEBVs by summing the allelic effects at each locus, using the posterior estimated SNP effects from the BayesR runs of the real dataset. 1000 simulations were run, each one using an estimate of SNP effect sizes from a different posterior sample of the BayesR MCMC chain used to perform the genomic prediction (i.e. the first gene-dropped dataset used estimated SNP effects from the 1st posterior sample of the MCMC chain, the second gene-dropped dataset used estimated SNP effects from the 2nd posterior sample, and so on). This ensures that uncertainty in the posterior estimate of each SNP effect size is carried into the gene-dropping simulations.

At the end of the gene-dropping process there were 1000 simulated datasets that could be compared to the 1000 MCMC samples of the real dataset. Thus, it was possible to determine whether observed changes in GEBVs over time were likely to be greater than can be expected from genetic drift alone. For each gene-dropped dataset we regressed the cohort mean GEBV on birth year weighting each year by sample size, exactly as in the real dataset. For each of the 1000 comparisons the slope of birth year from the gene-dropped dataset was subtracted from the slope of birth year in the real dataset, in order to generate a posterior distribution of temporal changes in GEBVs relative to those expected under drift. The probability of the observed microevolutionary change exceeding that expected from genetic drift was estimated as the proportion of the 1000 comparisons where the gene-dropped slope was greater than the real data slope.

For the pedigree-derived EBVs, the process of comparing observed trends with those expected under drift was the method advocated by Hadfield and adopted by others [35–37]. For each of the 1000 simulated datasets, the additive genetic variance was sampled from the posterior distribution of the MCMCglmm animal model. We used the phensim function of the R package pedantics v. 1.7 [40] to obtain simulated breeding values. Founders were assigned breeding values based on the additive genetic variance and progeny were assigned values by sampling from a Gaussian distribution with mean equal to the midparent breeding value and variance equal to half of the population additive genetic variance (for that sample of the posterior distribution of EBVs). Subsequent comparisons between the observed and simulated temporal trends in EBVs were performed exactly as for the genomic EBV analyses described above. Because we have used two different methods to simulate

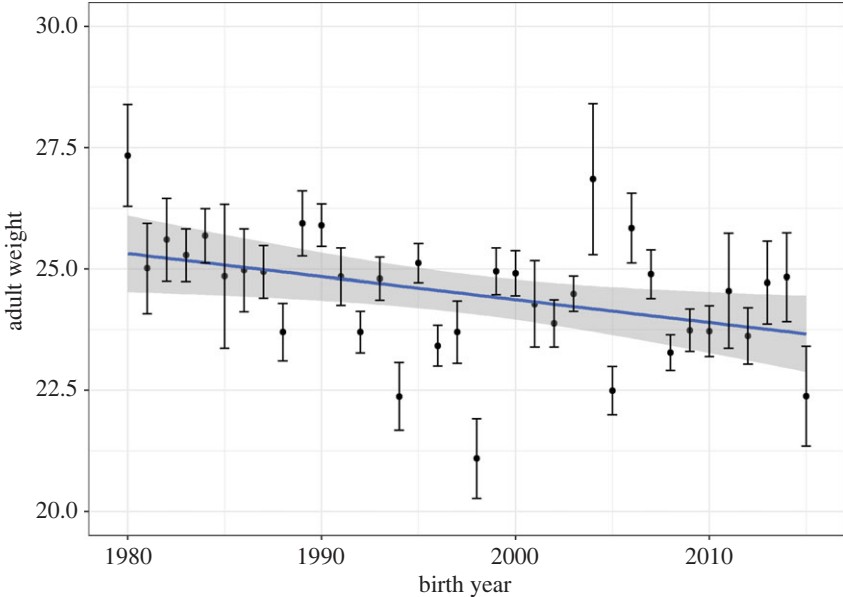

**Figure 1.** Adult weight (corrected for age and sex) has declined over the course of the long-term study. (Online version in colour.)

patterns of change expected by drift, we distinguish between statistical tests that employ these two simulated distributions with the notation $p_{\text{drift\_genomic}}$ and $p_{\text{drift\_pedigree}}$.

## (h) Genomic prediction using a 'leave one cohort out' approach

An additional genomic prediction analysis was run in which each cohort was treated as a test population (i.e. phenotypes unknown) and all of the other cohorts were treated as a training population. Thus, for every individual in the test cohort, the genomic prediction did not use the phenotypes of that individual or any other individual born in the same year. This ensures not only that the focal individual's phenotype does not contribute to its GEBV, but also reduces the risk that estimated SNP effect sizes are influenced by between-cohort-level covariances between allele frequencies and adult weight. The results support the main findings and are included in the electronic supplementary material (electronic supplementary material, table S6).

## 3. Results

### (a) Temporal changes in adult body weight

Adult body weight declined over the course of the study period (figure 1a), by an average of 0.067 kg per year ($p = 0.003$). However, in the period 2005–2015 (i.e. in the cohorts born since the Wilson et al. [18]), there has been a non-significant increase of 0.106 kg per year ($p = 0.23$; See electronic supplementary material).

### (b) Microevolutionary changes in body weight GEBVs

Between 1990 and 2015 mean cohort adult body weight GEBVs increased by approximately 0.011 (95% credible interval 0.001–0020) kg per year (figure 2a; electronic supplementary material, table S4). The posterior probability of no genetic change was low (probability = 0.014), using the conservative approach of regressing the posterior distribution of cohort mean GEBVs on year. A similar trend was seen with the pedigree-derived EBVs; an increase of 0.015 (95% credible interval 0.002–0.029) kg per year, $p = 0.013$

(figure 2b; electronic supplementary material, table S4). When comparing the posterior distribution of real cohort mean GEBVs regressed on year to those generated under gene-dropping genetic drift simulations, the distribution of the difference between real and gene-dropped regression coefficients was consistent with the observed trends being larger than expected from genetic drift (figure 2c,d; electronic supplementary material, table S7; probability that the GEBV slope is explainable by drift alone: $p_{\text{drift\_genomic}} = 0.057$, $p_{\text{drift\_pedigree}} = 0.028$; probability that the pedigree EBV slope is explainable by drift alone: $p_{\text{drift\_genomic}} = 0.042$, $p_{\text{drift\_pedigree}} = 0.022$). Thus, while genetic drift cannot be ruled out as an explanation for the observed increase in adult weight GEBVs, a response to selection for greater weights is perhaps more likely. In the electronic supplementary material we show that the main conclusions would be the same if all individuals from the 1980 cohort onwards are included, or if the analysis is restricted to only test population individuals from either 1980 or 1990 onwards (electronic supplementary material, tables S4 and S5, and figure S1).

During the period studied by Wilson and colleagues (1985–2005) the trends in GEBVs were similar to those for the longer 1990–2015 period (figure 2e). GEBVs increased by 0.012 (95% CI −0.002–0.026) kg per year and the posterior probability of no genetic change was low ($p = 0.042$). As with the extended dataset, the possibility that the observed genetic changes were attributable to genetic drift could not be excluded, although drift being the explanation seems reasonably unlikely ($p_{\text{drift\_genomic}} = 0.084$, $p_{\text{drift\_pedigree}} = 0.062$; electronic supplementary material, table S7; figure 2g). Pedigree-derived EBVs yield very similar findings (figure 2f,h; change in EBVs = 0.020 [CI = −0.000–0.041]; probability of no genetic change = 0.038; probability of genetic change being attributable to genetic drift: $p_{\text{drift\_genomic}} = 0.056$, $p_{\text{drift\_pedigree}} = 0.035$; (electronic supplementary material, tables S4 and S7).

Intriguingly, when the data are restricted to the period 2005–2015 the genomic and pedigree EBVs give qualitatively different patterns (electronic supplementary material, table S4 and figure S1). With the genomic EBVs, the trends are similar to those described for the longer 1980–2015, 1985–2005 and 1990–2015 time series (see electronic supplementary

*Proc. R. Soc. B* **289**: 20220330

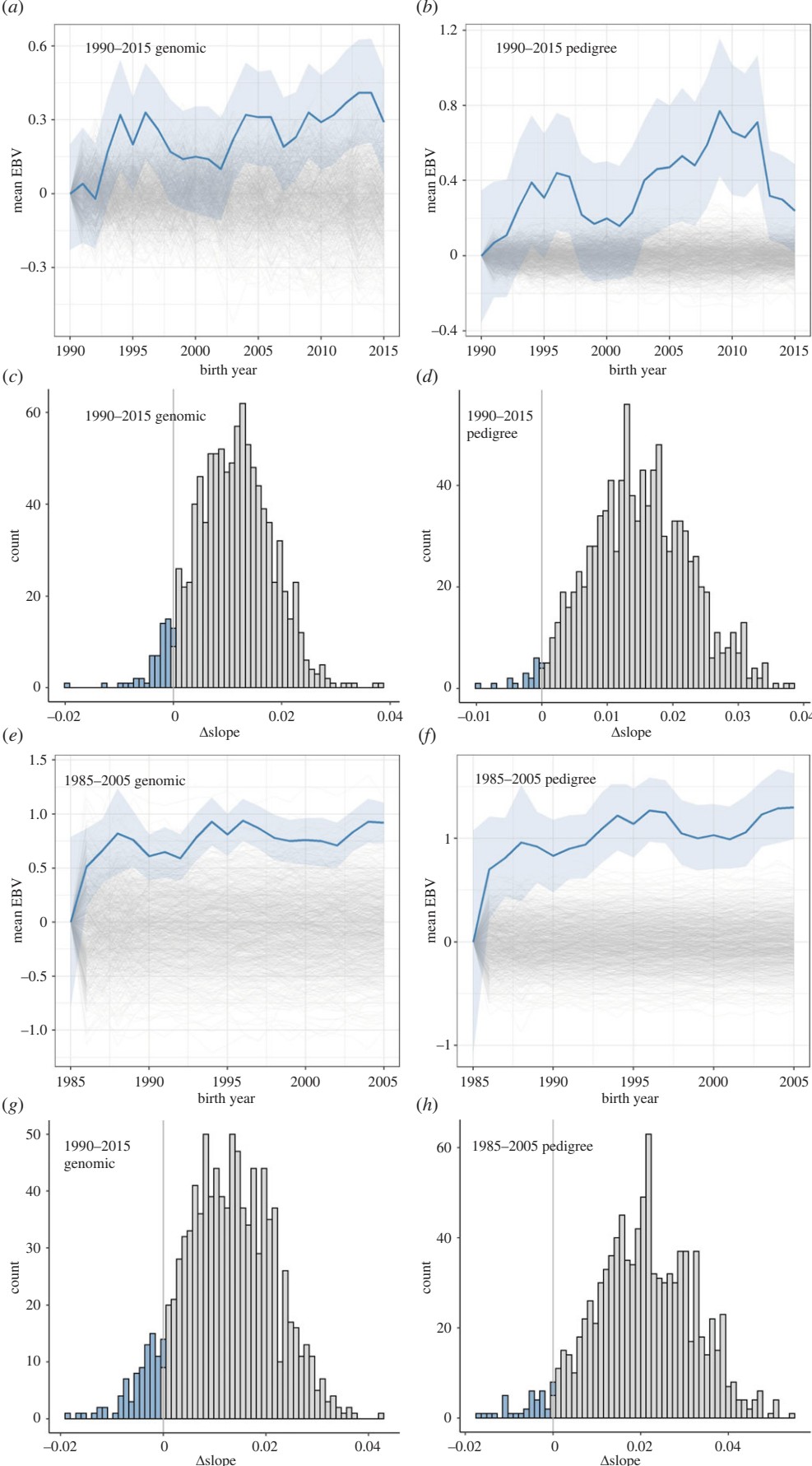

**Figure 2.** (*a*) GEBVs for adult birth weight have increased over the course of the study. Bold blue line is posterior cohort means. Light blue shading shows the 95% posterior credible interval. Thin grey lines are the 1000 gene-dropped simulations, showing the expected changes in adult weight GEBVs due to genetic drift. (*c*) The distribution of the difference in slope of GEBV against year for the real data minus the gene-dropped data. The proportion of slope differences less than 0 gives the probability that the observed slope is caused by genetic drift. Panels (*b*) and (*d*) show the equivalent results, using pedigreed-derived EBVs rather than genomic ones, and drift simulations using breeding values rather than the individual loci (see Methods). Data are from training and test population individuals from 1990–2015. Panels (*e*)–(*h*) show the same plots but for the period 1985–2005. (Online version in colour.)

material, tables S4 and S5). They are no longer significant, perhaps because the sample sizes are more modest; there are around half the number of animals as are in the longer time series. However, with the pedigree EBVs, the trend for 2005–2015 is negative (a decline in EBVs of approximately 0.023 kg per year) and is borderline significant (electronic supplementary material, table S4 and S5). In the electronic supplementary material, we provide a possible explanation of the discrepancy between the genomic and pedigree EBVs over this period (see 'Is the accuracy of EBV trends affected by pedigree depth?')

## 4. Discussion

The primary aim of this study was to investigate the use of genomic prediction as an alternative to a pedigree-based animal model approach to studying microevolutionary trends. There are two main advantages to a genomic prediction approach. First, it avoids the necessity of reconstructing a pedigree, which potentially opens the way to microevolutionary studies being conducted in a greater number of systems, including those with very large population sizes, long generation times or high rates of dispersal [41,42]. Second, some of the potential biases associated with pedigree-based approaches to studying microevolution can be avoided. For example, when a focal individual is part of a test population, its phenotype is not used to predict its GEBV, avoiding the problem that phenotyped individuals with low pedigree connectedness will tend to have GEBVs that reflect the environmental contribution to their phenotype [16]. Furthermore, simulations and empirical data have shown that the accuracy of an individual's genomic estimated breeding value is less sensitive to the number of relatives than when pedigree-based animal models are used [27]. Genomic approaches often outperform pedigree-based ones, including in studies of wild populations [25], but especially when close relatives are absent from the dataset [27]. There are further improvements that could be made to the approach used here. Most notably, there is not yet an easy-to-implement single-stage framework that for allows for Bayesian alphabet [43] genomic prediction models to be run when there are repeated measures and other random effects that might be fitted. Single-step genomic BLUP (GBLUP) methods are promising, as they can accommodate relationship matrices built jointly from SNPs and (if available) a pedigree, while also fitting non-genetic fixed and random effects [44]. However, we are not aware of current software that does this, while also outputting the sampled MCMC iterations which are required to assess the significance of the observed microevolutionary changes. In our dataset, there is little to be gained from constructing a relationship matrix that combines genotype and pedigree information as nearly all of the phenotyped animals are also genotyped. Here, we used a two-stage approach, running a mixed model to estimate phenotypes adjusted for non-genetic effects, before running the genomic prediction models. This does mean that uncertainty in the adjusted phenotypes are not carried through to the downstream analysis, although uncertainty in the underlying breeding values is hopefully accounted for. The pedigree-derived EBVs (which are from a single-stage model) and the GEBVs (which are not) produced similar conclusions. In fact, our conclusions from the GEBVs seem to be more conservative than those from the pedigree EBVs, but of course we cannot be certain that would be the case in other systems.

We extended a time series examining whether there was a microevolutionary response to selection for larger body weight in Soay sheep. Previous work had suggested breeding values for adult weight had increased [18], but subsequent scrutiny suggested the temporal change may be indistinguishable from stasis [17]. Here, the addition of 10 more years of data confirmed that breeding values have increased by approximately 0.010–0.015 kg per year. Furthermore, when analysing the same period as the previous study (1985–2005), the evidence for evolutionary change was now statistically significant, whether using genomic or pedigree-derived breeding values. A possible explanation is that the previous study suffered from the problems caused by EBVs in the later cohorts being harder to estimate accurately as those animals lacked phenotyped descendants. Here that problem is avoided by using either (i) GEBVs whose accuracy is not dependent on phenotyped descendants, or (ii) pedigree EBVs where phenotypic records were collected from another 10 years of descendants after the last cohort in the temporal trend analyses was born, meaning the accuracy of EBVs in the later cohorts was likely to be improved.

How does this rate of evolutionary change compare to other studies? After adding the GEBVs to the overall mean adult August body weight and log-transforming the data, we estimated adult body weight to be evolving at a rate of approximately 0.049 Haldanes (see electronic supplementary material). The largest published compilation of rates of microevolutionary change in wild populations [45,46], contains over 3000 estimates measured in Haldanes, albeit mostly estimated from phenotypic rather than genetic changes. Around 11% of those estimates exceed 0.049 Haldanes. Clearly, there will be relatively wide confidence intervals on our (and any other) estimate of rates of microevolution, but it seems that the rate observed here is relatively large. Thus, we should be cautious about whether genomic prediction approaches to studying microevolutionary change will be able to detect more modest responses to selection, especially as the accuracy of GEBVs measured in this population is relatively high [26], but that may not be the case in other systems. Given these considerations, we are reluctant to prescribe minimum sample sizes or number of markers required for similar investigations in other systems. However, we do recommend that researchers first establish the accuracy of GEBVs by cross-validation, or attempt to estimate the likely accuracy of GEBVs using formulae that predict the likely accuracy under given genetic architectures, genome sizes and effective population sizes [47].

Although, the earlier evidence for breeding values increasing was equivocal [17,18], adult weight at the phenotypic level was decreasing. However, this phenotypic decline seems to have been arrested and possibly reversed in the years since the earlier study. The causes of the possible reversal are unclear and are probably a complex combination of density, demographic and abiotic factors [20,48,49]. It should be noted that temporal trends in adult weight and in GEBVs between 2005–2015 are not significant, because the sample size is considerably smaller than that of the entire dataset. However, the regression slopes of cohort mean GEBVs on year are almost identical to those of the longer time series starting in 1980 or 1990. Thus, while the phenotypic trends for adult weight have probably changed

since the Wilson *et al.* [18] study, the genetic trends appear to have been more constant.

This is not the first study to have used simulations where breeding values are inherited through a pedigree to test whether observed microevolutionary trends are greater than expected by genetic drift [35–37]. However, other studies have used estimates of additive genetic variance from animal models to assign founder individuals a breeding value, under the assumption that the phenotype has a classical polygenic genetic architecture. Adult weight in Soay sheep is in fact a polygenic trait [21,29], but nonetheless the gene-dropping simulations explicitly modelled the number, effect size and genomic location of causal loci, using estimated SNP effects from each posterior sample of the genomic prediction analysis. Of course, the accuracy of the estimated SNP effects is unknown, but by using the posterior distribution of SNP effect sizes, the uncertainty in the estimates is incorporated into the drift simulations. Gene-dropping individual loci allow for variable effect sizes, and accommodates features that cannot be modelled by gene-dropping breeding values, such as the amount of linkage, recombination and LD between causal loci. Thus, gene-dropping simulations that explicitly model a trait's architecture should give a better reflection of what changes can occur due to drift, thereby making inferences about whether trends are due to selection more robust. In the electronic supplementary material we provide some evidence that simulations to test whether observed trends exceed expectations under drift are more conservative if the genetic architecture is explicitly modelled (see 'Comparison of two methods to simulate evolutionary change expected under drift' and table S7).

In summary, we have demonstrated a genomic approach to studying microevolutionary trends that should be robust and applicable to other systems. We show that genomic and pedigree-derived EBVs yield similar results, and where they do differ, known problems with pedigree-based methods are a plausible explanation. In the case of Soay sheep, there is convincing evidence that breeding values for adult weight have been increasing by around 0.01 kg per year over a period of more than 30 years, and that the trend has probably been driven by a response to selection for larger size rather than genetic drift. The rate of increase in breeding values has remained constant both during and after a period when phenotypic values were declining due to environmental or demographic effects.

Data accessibility. Data and scripts required to run the analyses can be found on Dryad Digital Repository: https://doi.org/10.5061/dryad.fj6q573sg [50].

The data are provided in electronic supplementary material [51].

Authors' contributions. D.C.H.: formal analysis, methodology, writing—review and editing; B.A.: formal analysis, methodology, writing—review and editing; C.B.: data curation, formal analysis, methodology, writing—review and editing; P.A.E.: data curation, formal analysis, methodology, writing—review and editing; S.E.J.: data curation, formal analysis, investigation, methodology, writing—review and editing; A.J.W.: conceptualization, writing—review and editing; J.G.P.: data curation, investigation, writing—review and editing; J.M.P.: conceptualization, data curation, funding acquisition, project administration, resources, writing—review and editing; J.S.: conceptualization, data curation, formal analysis, funding acquisition, investigation, methodology, project administration, supervision, writing—original draft.

All authors gave final approval for publication and agreed to be held accountable for the work performed therein.

Conflict of interest declaration. We declare we have no competing interests.

Funding. This work was funded by a Natural Environment Research Council (NERC) grant, (NE/M002896/1) awarded to J.S. and J.M.P. SNP genotyping was mostly funded by a European Research Council (ERC) grant (Wild Evolutionary Genomics) awarded to J.M.P.

Acknowledgements. We are grateful to The National Trust for Scotland (NTS) for allowing us to conduct fieldwork on St Kilda. QinetiQ and Eurest provided fieldwork logistical support. We are grateful to the many volunteers who have helped with field data collection on St Kilda over the last three decades. Genotyping was carried out at the Wellcome Trust Clinical Research Facility Genetics Core.

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
