## [Peer Review File · Proceedings of the Royal Society B: Biological Sciences]

Review History

RSPB-2020-2478.R0 (Original submission)

Review form: Reviewer 1

Recommendation

Reject – article is scientifically unsound

Scientific importance: Is the manuscript an original and important contribution to its field?

Good

General interest: Is the paper of sufficient general interest?

Marginal

Quality of the paper: Is the overall quality of the paper suitable?

Marginal

Is the length of the paper justified?

Yes

Should the paper be seen by a specialist statistical reviewer?

No

Do you have any concerns about statistical analyses in this paper? If so, please specify them explicitly in your report.

Yes

It is a condition of publication that authors make their supporting data, code and materials available - either as supplementary material or hosted in an external repository. Please rate, if applicable, the supporting data on the following criteria.

Is it accessible?

Yes

Is it clear?

Yes

Is it adequate?

No

Do you have any ethical concerns with this paper?

No

Comments to the Author

Dear Authors,

You here aimed to compare a previous pedigree-based approach to a quantitative genetic analysis in a wild population (Wilson et al. 2007 and Hadfield et al. 2010) with a genomic approach. Testing whether 'genomic' quantitative genetics outperforms 'classical' pedigree-based approaches is interesting and this seems to be a suitable data set for this.

My main concern with your analysis is that you did not really compare the earlier pedigree-based results with the 'genomic' results because you did not analyse the same period as Wilson et al. 2007 (namely 1985-2005). You here analysed only periods starting in 1980 and 1990 and ending in 2015, i.e. including the positive trend in phenotypes after 2005. I clearly see that ignoring newer data could lead to overlooking novel aspects (here the reversed phenotypic) trend. However, by analysing different periods than previously a meaningful comparison between the pedigree-based and genomic approach is impossible! You should either analyse here the original period or analyse the periods until 2015 with a pedigree-based model for comparison. Furthermore, I would like to note that none of the drift-corrected trends in Table S3 reaches formal statistical significance, which somewhat questions that the genomic approach outperforms the pedigree-based approach. Additionally, I would like to raise a number of issues with the statistical analysis, please see Specific Comments below.

This manuscript is very technical, which is obviously necessary because it addresses a fairly technical point. Nevertheless, I found the Introduction dwelled quite a bit on the pedigree-based approach and problems with BLUPs and could have introduced genomic prediction more thoroughly. Furthermore, I was quite surprised that you did not discuss the biology behind the reversed phenotypic trend with a single word. In general, I wondered whether this manuscript would not be too technical for PRSocB, even for a special issue.

specific comments:

L(ines)40-41: Fluctuating selection and heritability are not per se the issue here. Even if they vary, their average will still predict evolutionary change correctly. Problems arise if they covary with each other or other relevant eco-evolutionary parameters.

L42-44: Compare 'contergradient variation'. Might be good to mention this parallel here.

L79: I was initially confused by the term 'gene-dropping' because I assumed in each simulation a single 'gene' would be dropped 'from' the genome rather than 'through' the pedigree. Might be a non-native speaker thing but would it be possible to choose another term?

L123-125: As becomes evident later, this was not directly possible in your analysis!

L125-126: I think this still has to be shown.

L170: If I understood correctly, the trait analysed here is under selection. Would one hence not expect that causal loci could deviate from HW-equilibrium? Why not include these?

L181: Since the aim of this manuscript is to compare pedigree-based with genomic approaches, it would be instructive to report (or calculate) the accuracy of pedigree-based EBVs for comparison, see e.g. Gienapp et al. (2019).

L182-185: Why these choices for effect sizes and number of pseudo-observations? I am aware of analysis where effects sizes of SNPs could be either zero or non-zero but I found these four values a bit arbitrary. Why suddenly five pseudo-observations for the largest effect size?

L192-195: You here use BLUPs as phenotypes. I find this a bit surprising given the known issues with BLUPs. You discuss these related to EBVs but these limitations generally apply all BLUPs. It would be preferable to include fixed effects and random effects in the genomic prediction model. Alternatively, you could account for the prediction error in the BLUPs by using a Bayesian approach as you did when testing for temporal trends in GEBVs.

L258-261: Are really these estimates reported in the tables? If the trend in the real data set was about 0.01 and the difference between real data and 'gene dropped data' was also around 0.01, then the expected trend under drift would be around zero. In the original re-analysis (Hadfield et al. 2010) and similar papers the 'drift trend' was reported directly. So, I now wonder whether the expected change in breeding values after correcting for drift would be essentially zero!

L290-291: I do not agree here. First, no formal statistical significance is reached. Second, see the comment just above.

L300-302: This point has been made before, see e.g. Moore & Kukul (2002) or Gienapp et al. (2017). BTW, I found your manuscript generally lacking some citations to earlier relevant studies of genomic prediction or 'genomic quantitative genetics' in wild populations.

L311-314: This actually contradicts your statement in the Introduction on L123-125. I am not familiar with all software for genomic prediction but this is definitely possible when fitting a GRM in e.g. ASReml or MCMCglmm, i.e. following a so-called GBLUP approach.

L316-317: See above comment about using BLUPs from a Bayesian model.

L358: I am not really convinced by this evidence...

cited literature

Moore AJ, Kukul PF (2002) Quantitative genetic analysis of natural populations. *Nature Rev. Genet.* 3, 971-978 (2002).

P Gienapp, S Fior, F Guillaume, JR Lasky, VL Sork, K Csilléry (2017) Genomic quantitative genetics to study evolution in the wild. *Trends in Ecology and Evolution* 32: 897-908

P Gienapp, MPL Calus, VN Laine, ME Visser (2019) Genomic selection on breeding time in a wild bird population. *Evolution Letters* 3: 142-151

Review form: Reviewer 2

Recommendation

Major revision is needed (please make suggestions in comments)

Scientific importance: Is the manuscript an original and important contribution to its field?

Good

General interest: Is the paper of sufficient general interest?

Good

Quality of the paper: Is the overall quality of the paper suitable?

Marginal

Is the length of the paper justified?

No

Should the paper be seen by a specialist statistical reviewer?

No

Do you have any concerns about statistical analyses in this paper? If so, please specify them explicitly in your report.

Yes

It is a condition of publication that authors make their supporting data, code and materials available - either as supplementary material or hosted in an external repository. Please rate, if applicable, the supporting data on the following criteria.

Is it accessible?

Yes

Is it clear?

Yes

Is it adequate?

N/A

Do you have any ethical concerns with this paper?

No

Comments to the Author

In their study "Using genomic prediction to detect microevolutionary change of a quantitative trait" Hunter and colleagues investigate the microevolutionary responses to natural selection through genomic estimated breeding values for adult weight in a 35-year dataset of Soay sheep. While the empirical data showed decreased body size, genome derived breeding values suggest an increase in the trait. The authors further use gene dropping simulations to test whether the estimated change was the result of genetic drift. The authors conclude that the increase in genetic breeding values was larger than expected by drift and attribute this to cryptic microevolution. The overall idea of the study is interesting and addresses important questions in evolutionary genetics and the introduction, methods and discussion section are well written. However, the

extremely short results section does not include sufficient information to assess the results of the different analyses performed. In particular the discrepancy between observed phenotypic change and predicted change has to be thoroughly evaluated to come to the "cryptic microevolution" interpretation and exclude methodological artefacts as reason for opposing directions. A more in-depth analysis of the accuracy of genomic breeding values, comparison with previously estimated pedigree based breeding values and more information about the simulation results should be added.

Major comments:

- The study is highly dependent on accurate estimation of genomic estimated breeding values (gEBVs), however the accuracy of the estimates was not assessed in the current manuscript. It would be very useful to see prediction accuracies based on cross validation results.
- The methods describe different distributions for effect sizes in the gEBV estimation. how did these affect the outcome?
- I agree that eEBVs provide a promising path to study evolutionary trends, but given the statement that this study "introduced a genomic approach to studying microevolutionary trends", I would have expected a more detailed evaluation of the factors that influence the accuracy of the results. Probably including simulations evaluating the robustness. I think the authors have a great dataset on hand to address these questions.
- How many effect sizes were used/estimated? Can you maybe elaborate on the n (individuals) < p (markers) problem for effect size estimation and how that might influence your allele dropping results?
- It is not clear to me what the "test set" was used for in this study. This set cannot be used to assess the accuracy of the predictions.

Minor comments:

- The title suggests that "genomic prediction" has been used, but I think this is technically not the case as not the phenotype prediction was used but rather the effect size estimation.
- Please clearly state what data has been published before and what is unpublished. do I understand correctly that genotyping data for the full set is so far unpublished, or has it been used in other studies before?
- l. 323ff this should probably go into the results
- Supplementary figures and tables are not referenced in the text.
- is the correlation line in fig. 1 a based on year/means or individual values? Slope for females and males look much less severe when plotted from the supplied "Weight_Life.txt" data.
- in the supplementary file "Weight_Life.txt" non of the individuals has both a weight value and weight_gebv. How come? The correlation between those 2 would be interesting.

Decision letter (RSPB-2020-2478.R0)

17-Nov-2020

Dear Dr Slate:

I am writing to inform you that your manuscript RSPB-2020-2478 entitled "Using genomic prediction to detect microevolutionary change of a quantitative trait" has, in its current form, been rejected for publication in Proceedings B.

This action has been taken on the advice of referees, who have recommended that substantial revisions are necessary. With this in mind we would be happy to consider a resubmission, provided the comments of the referees are fully addressed. However please note that this is not a provisional acceptance.

Sincerely,
Professor Gary Carvalho
mailto: proceedingsb@royalsociety.org

Associate Editor
Board Member: 1
Comments to Author:
Dear Dr. Slate,

Thank you for your submission to the special issue on Wild Quantitative Genomics in Proceedings of the Royal Society B. Your manuscript has now been reviewed by 2 experts in the field. Overall, the reviewers were positive about your manuscript's research topic, questions, and dataset. The reviewers identified a number of concerns about the writing and the analysis. Many of these concerns could be addressed with rewriting the introduction to emphasize the broader appeal of the study and adding additional details about methods and results for clarity. However, addressing other comments will require some additional analyses. In particular, addressing both reviewers' requests to include comparisons of genomic breeding values and pedigree-based breeding values would be very helpful.

Reviewer(s)' Comments to Author:

Referee: 1

Comments to the Author(s)

Dear Authors,

You here aimed to compare a previous pedigree-based approach to a quantitative genetic analysis in a wild population (Wilson et al. 2007 and Hadfield et al. 2010) with a genomic approach. Testing whether 'genomic' quantitative genetics outperforms 'classical' pedigree-based approaches is interesting and this seems to be a suitable data set for this.

My main concern with your analysis is that you did not really compare the earlier pedigree-based results with the 'genomic' results because you did not analyse the same period as Wilson et al. 2007 (namely 1985-2005). You here analysed only periods starting in 1980 and 1990 and ending in 2015, i.e. including the positive trend in phenotypes after 2005. I clearly see that ignoring newer data could lead to overlooking novel aspects (here the reversed phenotypic) trend. However, by analysing different periods than previously a meaningful comparison between the pedigree-based and genomic approach is impossible! You should either analyse here the original period or analyse the periods until 2015 with a pedigree-based model for comparison. Furthermore, I would like to note that none of the drift-corrected trends in Table S3 reaches formal statistical significance, which somewhat questions that the genomic approach outperforms the pedigree-based approach. Additionally, I would like to raise a number of issues with the statistical analysis, please see Specific Comments below.

This manuscript is very technical, which is obviously necessary because it addresses a fairly technical point. Nevertheless, I found the Introduction dwelled quite a bit on the pedigree-based approach and problems with BLUPs and could have introduced genomic prediction more thoroughly. Furthermore, I was quite surprised that you did not discuss the biology behind the reversed phenotypic trend with a single word. In general, I wondered whether this manuscript would not be too technical for PRSocB, even for a special issue.

specific comments:

L(ines)40-41: Fluctuating selection and heritability are not per se the issue here. Even if they vary, their average will still predict evolutionary change correctly. Problems arise if they covary with each other or other relevant eco-evolutionary parameters.

L42-44: Compare 'contergradient variation'. Might be good to mention this parallel here.

L79: I was initially confused by the term 'gene-dropping' because I assumed in each simulation a single 'gene' would be dropped 'from' the genome rather than 'through' the pedigree. Might be a non-native speaker thing but would it be possible to choose another term?

L123-125: As becomes evident later, this was not directly possible in your analysis!

L125-126: I think this still has to be shown.

L170: If I understood correctly, the trait analysed here is under selection. Would one hence not expect that causal loci could deviate from HW-equilibrium? Why not include these?

L181: Since the aim of this manuscript is to compare pedigree-based with genomic approaches, it would be instructive to report (or calculate) the accuracy of pedigree-based EBVs for comparison, see e.g. Gienapp et al. (2019).

L182-185: Why these choices for effect sizes and number of pseudo-observations? I am aware of analysis where effects sizes of SNPs could be either zero or non-zero but I found these four values a bit arbitrary. Why suddenly five pseudo-observations for the largest effect size?

L192-195: You here use BLUPs as phenotypes. I find this a bit surprising given the known issues with BLUPs. You discuss these related to EBVs but these limitations generally apply all BLUPs. It would be preferable to include fixed effects and random effects in the genomic prediction model. Alternatively, you could account for the prediction error in the BLUPs by using a Bayesian approach as you did when testing for temporal trends in GEBVs.

L258-261: Are really these estimates reported in the tables? If the trend in the real data set was about 0.01 and the difference between real data and 'gene dropped data' was also around 0.01, then the expected trend under drift would be around zero. In the original re-analysis (Hadfield et al. 2010) and similar papers the 'drift trend' was reported directly. So, I now wonder whether the expected change in breeding values after correcting for drift would be essentially zero!

L290-291: I do not agree here. First, no formal statistical significance is reached. Second, see the comment just above.

L300-302: This point has been made before, see e.g. Moore & Kukul (2002) or Gienapp et al. (2017). BTW, I found your manuscript generally lacking some citations to earlier relevant studies of genomic prediction or 'genomic quantitative genetics' in wild populations.

L311-314: This actually contradicts your statement in the Introduction on L123-125. I am not familiar with all software for genomic prediction but this is definitely possible when fitting a GRM in e.g. ASReml or MCMCglmm, i.e. following a so-called GBLUP approach.

L316-317: See above comment about using BLUPs from a Bayesian model.

L358: I am not really convinced by this evidence...

cited literature

Moore AJ, Kukul PF (2002) Quantitative genetic analysis of natural populations. *Nature Rev. Genet.* 3, 971-978 (2002).

P Gienapp, S Fior, F Guillaume, JR Lasky, VL Sork, K Csilléry (2017) Genomic quantitative genetics to study evolution in the wild. *Trends in Ecology and Evolution* 32: 897-908

P Gienapp, MPL Calus, VN Laine, ME Visser (2019) Genomic selection on breeding time in a wild bird population. *Evolution Letters* 3: 142-151

Referee: 2

Comments to the Author(s)

In their study "Using genomic prediction to detect microevolutionary change of a quantitative trait" Hunter and colleagues investigate the microevolutionary responses to natural selection through genomic estimated breeding values for adult weight in a 35-year dataset of Soay sheep. While the empirical data showed decreased body size, genome derived breeding values suggest an increase in the trait. The authors further use gene dropping simulations to test whether the estimated change was the result of genetic drift. The authors conclude that the increase in genetic breeding values was larger than expected by drift and attribute this to cryptic microevolution. The overall idea of the study is interesting and addresses important questions in evolutionary genetics and the introduction, methods and discussion section are well written. However, the extremely short results section does not include sufficient information to assess the results of the different analyses performed. In particular the discrepancy between observed phenotypic change and predicted change has to be thoroughly evaluated to come to the "cryptic microevolution" interpretation and exclude methodological artefacts as reason for opposing directions. A more in-depth analysis of the accuracy of genomic breeding values, comparison with previously estimated pedigree based breeding values and more information about the simulation results should be added.

Major comments:

- The study is highly dependent on accurate estimation of genomic estimated breeding values (gEBVs), however the accuracy of the estimates was not assessed in the current manuscript. It would be very useful to see prediction accuracies based on cross validation results.
- The methods describe different distributions for effect sizes in the gEBV estimation. how did these affect the outcome?
- I agree that eEBVs provide a promising path to study evolutionary trends, but given the statement that this study "introduced a genomic approach to studying microevolutionary trends", I would have expected a more detailed evaluation of the factors that influence the accuracy of the results. Probably including simulations evaluating the robustness. I think the authors have a great dataset on hand to address these questions.
- How many effect sizes were used/estimated? Can you maybe elaborate on the n (individuals) $<$ p (markers) problem for effect size estimation and how that might influence your allele dropping results?
- It is not clear to me what the "test set" was used for in this study. This set cannot be used to assess the accuracy of the predictions.

Minor comments:

- The title suggests that "genomic prediction" has been used, but I think this is technically not the case as not the phenotype prediction was used but rather the effect size estimation.
- Please clearly state what data has been published before and what is unpublished. do I understand correctly that genotyping data for the full set is so far unpublished, or has it been used in other studies before?
- l. 323ff this should probably go into the results
- Supplementary figures and tables are not referenced in the text.
- is the correlation line in fig. 1 a based on year/means or individual values? Slope for females and males look much less severe when plotted from the supplied "Weight_Life.txt" data.
- in the supplementary file "Weight_Life.txt" non of the individuals has both a weight value and weight_gebv. How come? The correlation between those 2 would be interesting.

Author's Response to Decision Letter for (RSPB-2020-2478.R0)

See Appendix A.

RSPB-2021-2201.R0

Review form: Reviewer 1

Recommendation

Major revision is needed (please make suggestions in comments)

Scientific importance: Is the manuscript an original and important contribution to its field?

Good

General interest: Is the paper of sufficient general interest?

Acceptable

Quality of the paper: Is the overall quality of the paper suitable?

Good

Is the length of the paper justified?

Yes

Should the paper be seen by a specialist statistical reviewer?

No

Do you have any concerns about statistical analyses in this paper? If so, please specify them explicitly in your report.

No

It is a condition of publication that authors make their supporting data, code and materials available - either as supplementary material or hosted in an external repository. Please rate, if applicable, the supporting data on the following criteria.

Is it accessible?

Yes

Is it clear?

Yes

Is it adequate?

Yes

Do you have any ethical concerns with this paper?

No

Comments to the Author

Dear Authors,

My main concern with the previous version of your manuscript was the lacking comparison between pedigree-based and genomic quantitative genetic analyses. In this respect your manuscript has been considerably improved as both analyses are presented. As I wrote earlier ignoring new data and excluding them from the analysis is generally not a good idea. However, since your manuscript heavily refers back to the analysis of Wilson et al. (2007) it would be desirable if the same period than in this paper (1985-2005) would have been analysed with the genomic quantitative genetic analysis. Now, you present a pedigree-based analysis of a different period including the recent years (with the reversed phenotypic trend). While in the original analysis the genetic trend was not significant (after properly taking drift into account), it is now

significant in the pedigree-based analysis. It would make a far better point for the usefulness of genomic quantitative genetic approaches if the genomic approach could demonstrate a genetic trend while the pedigree-based approach could not. Actually, now the pedigree-based trend is significant while the genomic trend is not (although this may be a result of the more conservative gene-dropping test).

My other comments concerning the analyses (e.g. choice of prior, two-step analysis) have been satisfyingly addressed. However a few specific comments arose again:

L97 Ref 21 seems to be wrong in the list of references. Please double-check also all other references as I did not check all (I find numbered references generally cumbersome...).

L100-104 I am puzzled by this added explanation that refers to the recently reversed phenotypic trend, while this paragraph deals with the original paper by Wilson and later the re-analysis by Hadfield. To my understanding the contrasted phenotypic and genetic trends were thought to be caused by deteriorating environmental conditions. Should this addition not better be moved to L389-391 where you discuss this? The explanation given there differs from what you write here, though.

L415 'introduced' is a bit overstated as it implies that something conceptually novel was used.

L418/419 This evidence is partly only convincing because of the significant genetic trend from the pedigree-based analysis.

Review form: Reviewer 3

Recommendation

Accept with minor revision (please list in comments)

Scientific importance: Is the manuscript an original and important contribution to its field?

Good

General interest: Is the paper of sufficient general interest?

Good

Quality of the paper: Is the overall quality of the paper suitable?

Excellent

Is the length of the paper justified?

Yes

Should the paper be seen by a specialist statistical reviewer?

No

Do you have any concerns about statistical analyses in this paper? If so, please specify them explicitly in your report.

No

It is a condition of publication that authors make their supporting data, code and materials available - either as supplementary material or hosted in an external repository. Please rate, if applicable, the supporting data on the following criteria.

Is it accessible?

Yes

Is it clear?

Yes

Is it adequate?

Yes

Do you have any ethical concerns with this paper?

No

Comments to the Author

This ms explores the potential for using genomic prediction to detect microevolutionary trends using a long term study of Soay sheep.

I was not one of the original reviewers on this so have only looked at the revised ms and the authors response letters.

In my view the authors have done a great job in addressing all the comments from the reviewers in a very comprehensive manner and I think the ms is much improved as a result.

I just have a few small things:

Regarding the Haldane estimate provided in the discussion I am not sure this is correctly derived?

In Gingerich 1993 (that the authors cite) the log difference in mean is used, whereas here it is the difference in average (GEBV) between generations here (slope):

$rate(h) = (\ln x_2 / s_{\ln x}) - (\ln x_1 / s_{\ln x}) / t_2 - t_1$, where $\ln x_2$ is the natural log of the phenotype at time point 2 and $s_{\ln x}$ is the pooled SD of $\ln x_1$ s and $\ln x_2$ s.

The authors correctly take generation time into account to obtain a measure of change per generation but I think that this is not comparable to other estimates from the review article by Hendry et al. if they (presumably) used the log difference? Maybe worth to check up.

Also, Using the approach as authors on the phenotypic trend instead I get a Haldane estimate of $0.067 / 7.4^4 = 0.036$, which is pretty high..

My point is that I think the argument of genomic prediction being useful even at low intensities of selection does not necessarily follow from this calculation.

Throughout: perhaps it makes more sense to convert estimates of changes into grams instead of e.g. 0.01kg per year even if the slope of course is estimating kg/year. Given weight of the sheep we really are talking minuscule modest changes..

line 51: then not than

line 53: I would cite Hendersons original work as well , it was rediscovered by the wild quantitative group

line 207: I understood from the response to reviewers that it is very difficult to include uncertainty in the BLUPs into account with the current software available. Nonetheless, it is striking given what we know how ignoring uncertainty in EBVs caused a lot of wrong inferences to be made that, we are still in that situation with GEBVs.. I agree with the authors though that the added EBV analysis from MCMCGLMM go some way to support the inferences from GEBVs that ignore the uncertainty in BLUPs. One can only hope this is not specific to the situation here

but applies more generally...perhaps a statement to this effect can be added to where this is discussed (line 369-366).

line 214-215: maybe I am missing something but in the 'phenotype' for the GEBVs you include birth year and capture year as random effect whereas for the EBV analysis you use birth year as random effect but capture year as fixed effect? Why the difference?

line 228-229: so effectively a one-tailed test?

Line 348: but reconstructing a pedigree once you have all the molecular marker data is not really very difficult or time consuming so I dont really understand this first argument. I would say the potential to avoid biases is of much greater importance no?

line 397: what is a "largely polygenic trait"?

Line 414: so, 10g per year?

Decision letter (RSPB-2021-2201.R0)

10-Nov-2021

I am writing to inform you that this version of your manuscript RSPB-2021-2201 entitled "Using genomic prediction to detect microevolutionary change of a quantitative trait" has, in its current form, been rejected for publication in Proceedings B.

This action has been taken on the advice of referees, who have recommended that substantial revisions are necessary. With this in mind we would be happy to consider a resubmission, provided the comments of the referees are fully addressed. However please note that this is not a provisional acceptance.

Please find below the comments made by the referees, not including confidential reports to the Editor, which I hope you will find useful. Please note that it is highly unusual that we invite a second major resubmission. On this occasion, there is sufficient indication that your manuscript does contain potentially valuable and informative content, but currently, there are major changes required. In addition to the reanalysis, that I strongly endorse, plus the other comments made, there is one further concern that will require your careful consideration. While your manuscript represents potentially part of a special feature, it is vital that the accessibility of information and the generic impact, demonstrating a major advance in the field, is also fully satisfied. Therefore, while I appreciate that the topic is necessarily technical in nature, there are elements of your approach now buried within the supplementary information, and there is insufficient detail on how and why the generic value of your findings, will necessarily advance the state of art. Please do increase a little detail here, within your manuscript, especially in justification for the paper, in terms of addressing a key knowledge gap, or shortcoming, as well as its generic value

- 1) A 'response to referees' document including details of how you have responded to the comments, and the adjustments you have made.
- 2) A clean copy of the manuscript and one with 'tracked changes' indicating your 'response to referees' comments document.
- 3) Line numbers in your main document.
- 4) Please read our data sharing policies to ensure that you meet our requirements <https://royalsociety.org/journals/authors/author-guidelines/#data>.

Sincerely,
 Professor Gary Carvalho
 mailto: proceedingsb@royalsociety.org

Associate Editor Board Member

Comments to Author:

Overall the authors have done a great job incorporating suggestions from reviewers. However, the two reviewers raise a number of points that are important to address. The main remaining concern is that Reviewer 1 would like to see a reanalysis of the same time period considered in Wilson et al. 2007 and Hadfield et al 2010 (1985-2005). I recognize that addressing this comment may require substantial work; however I agree that the framing of the manuscript sets up the expectation that the same dataset will be reanalyzed. I think that directly comparing results to previous papers and then discussing the results after expanding the dataset would make the manuscript stronger.

I have a few additional comments:

For the tests of microevolutionary change, since pedigree-derived EBVs are simulated by assigning progeny midparental values, I wonder if it would be useful to use the same approach for the GEBV simulations and compare those results to the gene dropping results in the main text instead of the supplement. That way the main text could more directly compare tests of microevolutionary change using pedigree-derived EBVs vs GEBVs, and then compare tests using different ways of simulating expectations for GEBVs. The observed differences between pedigree and genomic EBVs may be due to both differences in the EBV estimates and the method used for simulating change under drift, and it would be nice to be able to disentangle the two.

For the discussion of pedigree depth, it would be helpful to add a figure showing the distribution of phenotyped close relatives for individuals of different birth cohorts.

If the primary aim of the paper is to introduce a genomic approach to studying microevolutionary change, it would be nice to include a paragraph in the discussion about requirements and caveats of such an approach (e.g., a discussion of minimum marker numbers and sample sizes, especially as it is difficult to accurately estimate small effect sizes with the sample sizes used in most studies of natural populations).

Lines 262-263: It is unclear why phased haplotypes were randomly assigned to founders in the gene dropping analyses; why not use the actual phased haplotypes for each founder?

Line 273: Why between 2 and 10 years? Is this the typical breeding lifespan of Soay sheep?

Reviewer(s)' Comments to Author:

Referee: 3

Comments to the Author(s).

This ms explores the potential for using genomic prediction to detect microevolutionary trends using a long term study of Soay sheep.

I was not one of the original reviewers on this so have only looked at the revised ms and the authors response letters.

In my view the authors have done a great job in addressing all the comments from the reviewers in a very comprehensive manner and I think the ms is much improved as a result.

I just have a few small things:

Regarding the Haldane estimate provided in the discussion I am not sure this is correctly derived?

In Gingerich 1993 (that the authors cite) the log difference in mean is used, whereas here it is the difference in average (GEBV) between generations here (slope):

$rate(h) = (\ln x_2 / s_{\ln x}) - (\ln x_1 / s_{\ln x}) / t_2 - t_1$, where $\ln x_2$ is the natural log of the phenotype at time point 2 and $s_{\ln x}$ is the pooled SD of $\ln x_1$ s and $\ln x_2$ s.

The authors correctly take generation time into account to obtain a measure of change per generation but I think that this is not comparable to other estimates from the review article by Hendry et al. if they (presumably) used the log difference? Maybe worth to check up.

Also, Using the approach as authors on the phenotypic trend instead I get a Haldane estimate of $0.067/7.4*4 = 0.036$, which is pretty high..

My point is that I think the argument of genomic prediction being useful even at low intensities of selection does not necessarily follow from this calculation.

Throughout: perhaps it makes more sense to convert estimates of changes into grams instead of e.g. 0.01kg per year even if the slope of course is estimating kg/year. Given weight of the sheep we really are talking minuscule modest changes..

line 51: then not than

line 53: I would cite Hendersons original work as well , it was rediscovered by the wild quantitative group

line 207: I understood from the response to reviewers that it is very difficult to include uncertainty in the BLUPs into account with the current software available. Nonetheless, it is striking given what we know how ignoring uncertainty in EBVs caused a lot of wrong inferences to be made that, we are still in that situation with GEBVs.. I agree with the authors though that the added EBV analysis from MCMCGLMM go some way to support the inferences from GEBVs that ignore the uncertainty in BLUPs. One can only hope this is not specific to the situation here but applies more generally...perhaps a statement to this effect can be added to where this is discussed (line 369-366).

line 214-215: maybe I am missing something but in the 'phenotype' for the GEBVs you include birth year and capture year as random effect whereas for the EBV analysis you use birth year as random effect but capture year as fixed effect? Why the difference?

line 228-229: so effectively a one-tailed test?

Line 348: but reconstructing a pedigree once you have all the molecular marker data is not really very difficult or time consuming so I dont really understand this first argument. I would say the potential to avoid biases is of much greater importance no?

line 397: what is a "largely polygenic trait"?

Line 414: so, 10g per year?

Referee: 1

Comments to the Author(s).

Dear Authors,

My main concern with the previous version of your manuscript was the lacking comparison between pedigree-based and genomic quantitative genetic analyses. In this respect your manuscript has been considerably improved as both analyses are presented. As I wrote earlier ignoring new data and excluding them from the analysis is generally not a good idea. However, since your manuscript heavily refers back to the analysis of Wilson et al. (2007) it would be desirable if the same period than in this paper (1985-2005) would have been analysed with the genomic quantitative genetic analysis. Now, you present a pedigree-based analysis of a different period including the recent years (with the reversed phenotypic trend). While in the original analysis the genetic trend was not significant (after properly taking drift into account), it is now significant in the pedigree-based analysis. It would make a far better point for the usefulness of genomic quantitative genetic approaches if the genomic approach could demonstrate a genetic trend while the pedigree-based approach could not. Actually, now the pedigree-based trend is significant while the genomic trend is not (although this may be a result of the more conservative gene-dropping test).

My other comments concerning the analyses (e.g. choice of prior, two-step analysis) have been satisfyingly addressed. However a few specific comments arose again:

L97 Ref 21 seems to be wrong in the list of references. Please double-check also all other references as I did not check all (I find numbered references generally cumbersome...).

L100-104 I am puzzled by this added explanation that refers to the recently reversed phenotypic trend, while this paragraph deals with the original paper by Wilson and later the re-analysis by Hadfield. To my understanding the contrasted phenotypic and genetic trends were thought to be caused by deteriorating environmental conditions. Should this addition not better be moved to L389-391 where you discuss this? The explanation given there differs from what you write here, though.

L415 'introduced' is a bit overstated as it implies that something conceptually novel was used.

L418/419 This evidence is partly only convincing because of the significant genetic trend from the pedigree-based analysis.

Author's Response to Decision Letter for (RSPB-2021-2201.R0)

See Appendix B.

RSPB-2022-0330.R0

Review form: Reviewer 4

Recommendation

Major revision is needed (please make suggestions in comments)

Scientific importance: Is the manuscript an original and important contribution to its field?

Good

General interest: Is the paper of sufficient general interest?

Good

Quality of the paper: Is the overall quality of the paper suitable?

Good

Is the length of the paper justified?

Yes

Should the paper be seen by a specialist statistical reviewer?

No

Do you have any concerns about statistical analyses in this paper? If so, please specify them explicitly in your report.

No

It is a condition of publication that authors make their supporting data, code and materials available - either as supplementary material or hosted in an external repository. Please rate, if applicable, the supporting data on the following criteria.

Is it accessible?

Yes

Is it clear?

Yes

Is it adequate?

Yes

Do you have any ethical concerns with this paper?

No

Comments to the Author

Using genomic prediction to detect microevolutionary change of a quantitative trait, by Hunter and coworkers. The authors revisit the testing of the existence of microevolution for adult weight in the well-known Soay sheep population by comparing the evolution of observed breeding values with the ones simulated under the assumption of only drift affecting the performance for the trait. The relevant novelty is the use of genomic estimates of individual values instead of the pedigree-based EBVs. The use of genomic information for this task presents clear advantages from the past methodologies improving the accuracy of estimates and enlarging the type of populations that can be targets for such kind of studies.

Notwithstanding, I have a couple of concerns on the methodology used for the study. Firstly, I think that in the particular scenario of wild populations (and many times in domestic ones) the genomic evaluation (i.e. the estimation of the GEBVs) could be better under a GBLUP framework

instead of the SNP-BLUP methodology. The strategy of calculating the relationship matrix from the (mostly) genomic information and then applying classical mixed models would allow for a better use of available information. If relying exclusively on genomic data, GBLUP and SNP-BLUP methods have been proven to be equivalent. Moreover, using the single-step approach (a revision in, for example, Lourenco et al. 2002, Genes and reference therein) makes possible to include the information from all kind of individual, whatever they are genotyped or not but connected by genealogy. Genomic data allows for a more accurate estimation of actual/observed genetic relationships and a better use of within family variation (which is sometimes not possible using pedigrees). GBLUP methodology also allows for the use of more complex models like including maternal effects or repeated measures. If there is a sound sign of the trait being controlled by loci explaining different percentages of the genetic variance, this can be accounted for when constructing the GRM matrix by weighting differentially the information from each marker (see, for example, Tiezzi and Maltecca 2015, GSE).

Secondly, I agree with the authors that in the simulation step it is better to take an approach where individual SNPs are simulated, especially when the trait is controlled by few loci of large effects. However, I wonder how robust this methodology is against departures from the real 'genetic architecture'. If the number and effects of the causal loci are poorly estimated, maybe this would lead to wrong conclusions and assuming pure infinitesimal could be a better (more conservative) option. Realise that, besides the possible major genes, most quantitative trait also present a polygenic background.

I think that, at least, the influence of both items should be discussed.

Further comments follow:

- Page 3, lines 48-49: This process is usually referred to as calculating the 'genetic gain' in the context of breeding programs.

- Page 7, lines 140-141: As said before, the knowledge of the number and effect of the loci affecting a trait is not trivial.

- Page 8, line 157: To what extent is the sampling complete?

- Page 8, lines 158-159: Is mortality measured from the discovering of dead animals (frequent/extensive search in the field)? Or by the 'absence' of known animals in the annual control?

- Page 8, line 163: Even if you do not go for a in depth explanation on the way data are obtained, at least the reader should be warned/recalled that raw data are pre-corrected by some factors like sex and birth weight.

- Page 8, line 169: These are realisations that weight at different ages are really different traits (i.e. with different genetic control) and, thus, can not be included in the same evaluation.

- Page 9, lines 180 and 182: Some units are required (percentages).

- Page 10, lines 202-204: What about non genotyped (either with phenotype or not) but genealogically connected?

- Page 10, line 206: Did you try to include age as a covariate? Was the reproductive status of ewes accounted for in the evaluation?

- Page 10, line 215: Is permanent environment effect absent from the model for genomic evaluations?

- Page 10, lines 217-219: Include some explanation on the need of different number of total and burn-in iterations in the pedigree based and genomic evaluations. Link then with the gap between sampled iterations, to end up with the same number of samples.

- Page 12, line 252: Usually, beside the midparental breeding value, BVs of offspring includes a random value reflecting the Mendelian sampling, depending on the additive genetic variance for the trait and the inbreeding level of the parents. Was this term not included in the cited studies?

- Page 12, line 261: Please, provide the rationale/argumentation for such figure.

- Page 16, line 333: A space is missing before `Pdrift_genomic`.

- Page 17, line 341: But in Fig. 2e it can be seen that most of the increase is in the early years. This should be pointed out and included in the discussion of the results.

- Page 17, line 351: This is also observable in Figure 2b too.

- Page 17, lines 357-359: It is a bit awkward saying that you will present (relevant) explanations on the possible causes of the performance in the Supplementary Material. I guess that the concept

should be included in the main text and, if a formal prove exists, develop it in the Supplementary Material.

- Page 18, line 365: I agree that some wild population (even some domestic ones) can not be studied with a pedigree approach because it can be poor (scarce and not reliable). However, it has to be realised that the budget for such populations uses to be limited. Any comment on the feasibility of conducting large-scale genomic studies in this kind of populations.
- Page 18, lines 374-375: But genomic evaluation loses accuracy the lower is the relationship between the training and the testing population.
- Page 19, line 392: 'Surprisingly' is, for me, a too strong word. Reword the sentence to say something like 'Contrarily to what was found in previous studies ...'.
- Page 20, line 412: Noun (rate) and verb (are) do not match.
- Page 20, line 417: An extra comma.
- Page 20, lines 417-418: Do you think that this could be clarified/determined by extensive simulation studies?
- Page 21, line 440: Said before, some consequences on the inaccurate estimation of number of loci and their effects.
- Page 21, lines 452-454: I do not totally agree. In Fig. 2e it seem that EBVs stabilise from 1995.

Decision letter (RSPB-2022-0330.R0)

16-Mar-2022

Dear Dr Slate:

Your manuscript has now been peer reviewed and the reviews have been assessed by an Associate Editor. The reviewers' comments (not including confidential comments to the Editor) and the comments from the Associate Editor are included at the end of this email for your reference. As you will see, the reviewers and the Editors have raised some concerns with your manuscript and we would like to invite you to revise your manuscript to address them.

Research ethics:

Use of animals and field studies:

It is a condition of publication that you make available the data and research materials supporting the results in the article (<https://royalsociety.org/journals/authors/author-guidelines/#data>). Datasets should be deposited in an appropriate publicly available repository and details of the associated accession number, link or DOI to the datasets must be included in the Data Accessibility section of the article (<https://royalsociety.org/journals/ethics-policies/data-sharing-mining/>). Reference(s) to datasets should also be included in the reference list of the article with DOIs (where available).

Please submit a copy of your revised paper within three weeks. If we do not hear from you within this time your manuscript will be rejected. If you are unable to meet this deadline please let us know as soon as possible, as we may be able to grant a short extension.

Best wishes,
 Professor Gary Carvalho
 mailto: proceedingsb@royalsociety.org

Associate Editor Board Member

Comments to Author:

The authors have done a thorough job of addressing previous suggestions, and I think this manuscript will be a valuable contribution to our special feature. This manuscript has also been evaluated by an expert reviewer, who has two methodological concerns and a number of suggestions for additional clarification. Their detailed comments seem addressable, and I think it would be acceptable to not redo/add analyses and instead just add an acknowledgment of other methods for estimating GEBVs and a brief discussion of the robustness of the gene dropping simulations to errors in genetic architecture.

I have just a few small additional comments:

Lines 417-418: This sentence does not seem particularly helpful. While I agree with the authors that minimum sample sizes and marker numbers are highly dependent on the study system, I wonder if it would be more useful (for readers) to replace this sentence with a suggestion that folks run simulations of traits with different genetic architectures and varying levels of LD to help assess sample requirements.

Tables S4 & S5: Might be good to use the same terminology as the main text (P_drift_genomic / P_drift_pedigree)

Typo in Figure S1 caption: "1990-2105"

Reviewer(s)' Comments to Author:

Referee: 4

Comments to the Author(s).

Using genomic prediction to detect microevolutionary change of a quantitative trait, by Hunter and coworkers. The authors revisit the testing of the existence of microevolution for adult weight in the well-known Soay sheep population by comparing the evolution of observed breeding values with the ones simulated under the assumption of only drift affecting the performance for the trait. The relevant novelty is the use of genomic estimates of individual values instead of the pedigree-based EBVs. The use of genomic information for this task presents clear advantages from the past methodologies improving the accuracy of estimates and enlarging the type of populations that can be targets for such kind of studies.

Notwithstanding, I have a couple of concerns on the methodology used for the study. Firstly, I think that in the particular scenario of wild populations (and many times in domestic ones) the genomic evaluation (i.e. the estimation of the GEBVs) could be better under a GBLUP framework instead of the SNP-BLUP methodology. The strategy of calculating the relationship matrix from the (mostly) genomic information and then applying classical mixed models would allow for a better use of available information. If relying exclusively on genomic data, GBLUP and SNP-BLUP methods have been proven to be equivalent. Moreover, using the single-step approach (a revision in, for example, Lourenco et al. 2002, Genes and reference therein) makes possible to include the information from all kind of individual, whatever they are genotyped or not but connected by genealogy. Genomic data allows for a more accurate estimation of actual/observed genetic relationships and a better use of within family variation (which is sometimes not possible using pedigrees). GBLUP methodology also allows for the use of more complex models like including maternal effects or repeated measures. If there is a sound sign of the trait being controlled by loci explaining different percentages of the genetic variance, this can be accounted for when constructing the GRM matrix by weighting differentially the information from each marker (see, for example, Tiezzi and Maltecca 2015, GSE).

Secondly, I agree with the authors that in the simulation step it is better to take an approach where individual SNPs are simulated, especially when the trait is controlled by few loci of large

effects. However, I wonder how robust this methodology is against departures from the real 'genetic architecture'. If the number and effects of the causal loci are poorly estimated, maybe this would lead to wrong conclusions and assuming pure infinitesimal could be a better (more conservative) option. Realise that, besides the possible major genes, most quantitative trait also present a polygenic background.

I think that, at least, the influence of both items should be discussed.

Further comments follow:

- Page 3, lines 48-49: This process is usually referred to as calculating the 'genetic gain' in the context of breeding programs.
- Page 7, lines 140-141: As said before, the knowledge of the number and effect of the loci affecting a trait is not trivial.
- Page 8, line 157: To what extent is the sampling complete?
- Page 8, lines 158-159: Is mortality measured from the discovering of dead animals (frequent/extensive search in the field)? Or by the 'absence' of known animals in the annual control?
- Page 8, line 163: Even if you do not go for a in depth explanation on the way data are obtained, at least the reader should be warned/recalled that raw data are pre-corrected by some factors like sex and birth weight.
- Page 8, line 169: These are realisations that weight at different ages are really different traits (i.e. with different genetic control) and, thus, can not be included in the same evaluation.
- Page 9, lines 180 and 182: Some units are required (percentages).
- Page 10, lines 202-204: What about non genotyped (either with phenotype or not) but genealogically connected?
- Page 10, line 206: Did you try to include age as a covariate? Was the reproductive status of ewes accounted for in the evaluation?
- Page 10, line 215: Is permanent environment effect absent from the model for genomic evaluations?
- Page 10, lines 217-219: Include some explanation on the need of different number of total and burn-in iterations in the pedigree based and genomic evaluations. Link then with the gap between sampled iterations, to end up with the same number of samples.
- Page 12, line 252: Usually, beside the midparental breeding value, BVs of offspring includes a random value reflecting the Mendelian sampling, depending on the additive genetic variance for the trait and the inbreeding level of the parents. Was this term not included in the cited studies?
- Page 12, line 261: Please, provide the rationale/argumentation for such figure.
- Page 16, line 333: A space is missing before Pdrift_genomic.
- Page 17, line 341: But in Fig. 2e it can be seen that most of the increase is in the early years. This should be pointed out and included in the discussion of the results.
- Page 17, line 351: This is also observable in Figure 2b too.
- Page 17, lines 357-359: It is a bit awkward saying that you will present (relevant) explanations on the possible causes of the performance in the Supplementary Material. I guess that the concept should be included in the main text and, if a formal prove exists, develop it in the Supplementary Material.
- Page 18, line 365: I agree that some wild population (even some domestic ones) can not be studied with a pedigree approach because it can be poor (scarce and not reliable). However, it has to be realised that the budget for such populations uses to be limited. Any comment on the feasibility of conducting large-scale genomic studies in this kind of populations.
- Page 18, lines 374-375: But genomic evaluation loses accuracy the lower is the relationship between the training and the testing population.
- Page 19, line 392: 'Surprisingly' is, for me, a too strong word. Reword the sentence to say something like 'Contrarily to what was found in previous studies ...'.
- Page 20, line 412: Noun (rate) and verb (are) do not match.
- Page 20, line 417: An extra comma.
- Page 20, lines 417-418: Do you think that this could be clarified/determined by extensive simulation studies?
- Page 21, line 440: Said before, some consequences on the inaccurate estimation of number of loci and their effects.

- Page 21, lines 452-454: I do not totally agree. In Fig. 2e it seem that EBVs stabilise from 1995.

Author's Response to Decision Letter for (RSPB-2022-0330.R0)

See Appendix C.

Decision letter (RSPB-2022-0330.R1)

12-Apr-2022

Dear Dr Slate

I am pleased to inform you that your manuscript entitled "Using genomic prediction to detect microevolutionary change of a quantitative trait" has been accepted for publication in Proceedings B.

Data Accessibility section

Open Access

Paper charges

All supplementary materials accompanying an accepted article will be treated as in their final form. They will be published alongside the paper on the journal website and posted on the online

figshare repository. Files on figshare will be made available approximately one week before the accompanying article so that the supplementary material can be attributed a unique DOI.

Sincerely,
Professor Gary Carvalho
Editor, Proceedings B
mailto: proceedingsb@royalsociety.org

Associate Editor:

Board Member

Comments to Author:

The authors have done a good job addressing reviewer comments. Thanks for a nice contribution to our special issue!

The
University
Of
Sheffield.

School
Of
Biosciences

Prof Gary Carvalho
c/o Shalene Singh-Shepherd
Editorial Office
Proceedings B

5th October 2021

Professor Jon Slate
School of Biosciences
Alfred Denny Building (Room B78)
University of Sheffield
Western Bank
Telephone: +44 (0) 114 222 0048
Email: j.slate@sheffield.ac.uk

Dear Prof Carvalho / Senior Editor Singh-Shepherd

Thank you for giving us the opportunity to submit a revision of our manuscript *Using genomic prediction to detect microevolutionary change of a quantitative trait*. We are most grateful for the deadline extension, as the last year has proven particularly challenging to focus on research and outstanding manuscript revisions. The additional time granted to us has, we hope, enabled us to do justice to the helpful and constructive comments from the handling editor and the reviewers. Below we respond to each of the reviewers' points – our responses are in blue, italicised font. We have also submitted a clean and marked-up version of the manuscript, a revised supplementary information document and (for the reviewers, but not publication) a copy of a manuscript that has been provisionally accepted at *Molecular Ecology*. This manuscript was submitted to the bioRxiv preprint server at the time we originally submitted the paper, but although we cited it, we did not upload it with our submission. In hindsight this was a mistake, as some of the questions raised by the reviewers were directly addressed in that paper. We have undertaken some substantial new analyses in this revised manuscript and have therefore updated the files and code on the Dryad submission linked to this manuscript. We very much hope that the editors and reviewers agree with us that the manuscript is much improved on the earlier submission.

Your sincerely

Jon Slate

Associate Editor

Board Member: 1

Comments to Author:

Dear Dr. Slate,

Thank you for your submission to the special issue on Wild Quantitative Genomics in Proceedings of the Royal Society B. Your manuscript has now been reviewed by 2 experts in the field. Overall, the reviewers were positive about your manuscript's research topic, questions, and dataset. The reviewers identified a number of concerns about the writing and the analysis. Many of these concerns could be addressed with rewriting the introduction to emphasize the broader appeal of the study and adding additional details about methods and results for clarity. However, addressing other comments will require some additional analyses. In particular, addressing both reviewers' requests to include comparisons of genomic breeding values and pedigree-based breeding values would be very helpful.

We thank the reviewers and Associate Editor for their constructive and helpful comments. We have tried to address all of their points and in particular we have now conducted some additional analyses that compare results when using genomic estimated breeding values (GEBVs) and pedigree estimated breeding values. The conclusions are very consistent regardless of the method used, although the GEBVs are perhaps slightly more conservative. In particular, the evidence that evolutionary change is greater than expected from genetic drift alone is stronger when using pedigree EBVs. Below, we outline, point-by-point, how we have addressed the comments. One important point to note is that we could have done a better job of making the authors aware of a related manuscript that explores the accuracy of our genomic prediction for this trait and 7 other traits. While we did cite the preprint of that manuscript, we did not upload it, and as a result one of the reviewers has quite a lot of questions about the accuracy of our genomic prediction. That other manuscript has now been accepted for publication in Molecular Ecology, pending minor revisions, which are well underway. We have now uploaded the other manuscript and its supplementary information, as we feel it answers some of the more technical questions raised by the reviewers, especially Reviewer 2.

As a consequence of estimating and analysing pedigree-derived estimated breeding values the main points are:

- 1) *Pedigree and Genomic EBVs give very similar patterns across the overall dataset, with the evidence for changes in weight breeding values being due to selection rather than drift being marginally strengthened.*
- 2) *At the end of the time series, pedigree and genomic EBVs do give different results. We discuss these differences, and in particular, perform some new analyses that suggest this difference is likely to be due to the pedigree-EBVs being less reliable than the genomic ones.*
- 3) *We compare two methods that perform simulations of changes due to genetic drift – the established method where breeding values are transmitted down a pedigree, and the method introduced in this paper, where individual loci with estimated effect sizes are transmitted down the pedigree and their allelic effects summed in each individual. The latter approach explicitly models the trait's architecture rather than assumes a pseudo-infinitesimal model of allelic effects, and it appears to be slightly more conservative.*

Reviewer(s)' Comments to Author:

Referee: 1

Comments to the Author(s)

Dear Authors,

You here aimed to compare a previous pedigree-based approach to a quantitative genetic analysis in a wild population (Wilson et al. 2007 and Hadfield et al. 2010) with a genomic approach. Testing whether 'genomic' quantitative genetics outperforms 'classical' pedigree-based approaches is interesting and this seems to be a suitable data set for this.

We are pleased to see that the reviewer thinks the question is interesting and the data are appropriate. The comments they have raised (below) are valid and insightful. It was perhaps not our original intention to provide a formal comparison between genomic and pedigree-based approaches, but clearly it would have been better to do so, and we now have now made this comparison. We have also endeavoured to address the remaining comments.

My main concern with your analysis is that you did not really compare the earlier pedigree-based results with the 'genomic' results because you did not analyse the same period as Wilson et al. 2007 (namely 1985-2005). You here analysed only periods starting in 1980 and 1990 and ending in 2015, i.e. including the positive trend in phenotypes after 2005. I clearly see that ignoring newer data could lead to overlooking novel aspects (here the reversed phenotypic) trend. However, by analysing different periods than previously a meaningful comparison between the pedigree-based and genomic approach is impossible! You should either analyse here the original period or analyse the periods until 2015 with a pedigree-based model for comparison. Furthermore, I would like to note that none of the drift-corrected trends in Table S3 reaches formal statistical significance, which somewhat questions that the genomic approach outperforms the pedigree-based approach. Additionally, I would like to raise a number of issues with the statistical analysis, please see Specific Comments below.

We have now performed formal comparisons between the genomic and pedigree-based approaches to generate EBVs. We have done this by using the pedigree to fit a Bayesian animal model using the MCMCglmm package (Hadfield, 2010). A Bayesian method is appropriate because (i) this is the approach advocated by Hadfield and colleagues (Hadfield et al., 2010) when they introduced many of the problems and solutions associated with detecting evolutionary trends and (ii) it means patterns detected by the genomic and pedigree EBVs are directly comparable as both provide

uncertainty around the individual EBVs. We have used the period 1980-2015, as recommended by the reviewer (as well as the shorter 1990-2015 and 2005-2015 windows).

Our main finding, that there is a temporal increase in breeding values of weight, which is probably greater than can be expected by genetic drift alone, seems to be supported by the new analyses; the coefficients of EBVs over time are greater and more strongly significant than when we used genomic EBVs. However, we also noticed that the 'traditional' method of simulating changes by drift, whereby breeding values are assigned to founders and their progeny are assigned midparental values, is slightly less conservative than the approach we used here, which drops alleles at causal loci down the pedigree, following rules of Mendelian inheritance and recombination. We discuss this observation in the supplementary material, as it is not central to the main findings, but we think it will interest researchers studying microevolution of quantitative traits.

One other observation is perhaps noteworthy. The correlation between genomic and pedigree EBVs is very high (see next page) although there is an interesting pattern in the last three years of the dataset (2013-2015) when the variance in pedigree EBVs is lower than in other years (and the mean is centred around ~ 0), and as a result the slope of GEBVs (x-axis) against pedigree EBVs (y-axis) is flatter than other years. We think it is likely that this is due to a phenomenon described by Postma (2005) where individuals at the end of the pedigree have few, if any, descendants and as a result their EBV may be less reliable (Postma, 2006). Here it seems that there may be a tendency for these individuals to have EBVs closer to zero (i.e. the EBVs of the base population). The pattern is seen with individuals whether or not the focal individuals have been phenotyped, so it does not appear to be driven by the focal individual's EBV being largely determined by its own phenotype. Genomic EBVs are not sensitive to the same issue because they do not use pedigree information and the SNP effects they use to estimate GEBVs come from the entire population, not just relatives.

Table 1: Pedigree EBVs plotted against genomic EBVs, one cohort per panel.

This manuscript is very technical, which is obviously necessary because it addresses a fairly technical point. Nevertheless, I found the Introduction dwelled quite a bit on the pedigree-based approach and problems with BLUPs and could have introduced genomic prediction more thoroughly. Furthermore, I was quite surprised that you did not discuss the biology behind the reversed phenotypic trend with a single word. In general, I wondered whether this manuscript would not be too technical for PRSocB, even for a special issue.

We deliberately did not spend too much time introducing genomic prediction more generally because (i) our manuscript in review at Molecular Ecology describes the background and technical details in much greater detail and we wanted to avoid repetition of that information. We have uploaded that manuscript for reviewers (previously it was referenced as a bioRxiv preprint). (ii) There is another article in this special issue that reviews the potential and application of genomic prediction in wild populations (McGaugh et al., 2021), which we now cite, along with some other relevant empirical papers.

The reversed phenotypic trend (i.e. animal weights getting lighter in recent years) has been described previously and is not really the focus of this paper (other than that trend to be necessary for our trends in breeding values to show cryptic microevolution rather than non-cryptic microevolution). However, we do now expand that section, summarising the findings of a paper by Ozgul and colleagues, which showed that in the latter years of the study relatively light sheep that once would have died are now surviving. This results in a reduction in the overall mean adult weight. The improved survival is thought to be due to harsh winters becoming rarer (Ozgul et al., 2009).

specific comments:

L(ines)40-41: Fluctuating selection and heritability are not per se the issue here. Even if they vary, their average will still predict evolutionary change correctly. Problems arise if they covary with each other or other relevant eco-evolutionary parameters.

Thank you – we have now corrected this statement to read ‘fluctuating environmental conditions covarying with either....’.

L42-44: Compare ‘contergradient variation’. Might be good to mention this parallel here.

We did consider this as there are obvious parallels with the cryptic microevolution we describe, but countergradient variation is most often used when considering two (or more) populations inhabiting a gradient of environmental conditions. That is not something we are really looking at here, and given the space constraints, felt it was best not to expand this section any further.

L79: I was initially confused by the term ‘gene-dropping’ because I assumed in each simulation a single ‘gene’ would be dropped ‘from’ the genome rather than ‘through’ the pedigree. Might be a non-native speaker thing but would it be possible to choose another term?

We assumed that Gene-dropping was a reasonably familiar term for this type of analysis, when simulating the inheritance of alleles at multiple independently segregating loci (or indeed linked loci). We think the term was first coined in a conservation genetics study in 1986 (MacCluer et al.,

1986), but it is more commonly used by the human genetics community e.g. (Abecasis et al., 2002). Nonetheless, we have added a brief explanation in lines xxx, because if the reviewer is not familiar with the term, other readers will probably also be unfamiliar with it. In hindsight, the term is less widely used than we appreciated, although we do think it is a good description of the process of simulating haplotypes in founders and then assigning alleles to their descendants following Mendelian rules and recombination. We have removed the term 'gene-dropping' when describing the similar process of assigning breeding values to founders and midparental breeding values to their offspring; the process that has frequently been employed to estimate the changes in breeding values that might be expected through drift alone, such as in the appendix of (Hadfield et al., 2010).

We write:

“Third, we use a ‘gene-dropping’ simulation approach, to formally test whether any temporal patterns are likely to be due to a response to selection or are explainable by genetic drift. These simulations are analogous to methods that simulate the inheritance of breeding values down a pedigree. However, because the genomic prediction method estimates the contribution of each SNP to variation in weight, the genetic drift simulations are explicitly based on a description of the trait’s genetic architecture (the number and effect size of causal loci) rather than assuming an infinitesimal model. Here, haplotypes of causal loci are assigned to founders, and offspring inherit alleles at each locus according to Mendelian inheritance and empirically estimated recombination rates between linked loci”.

L123-125: As becomes evident later, this was not directly possible in your analysis!

We do adjust our phenotypes for fixed and random effects (sex, age, birth year, capture year and permanent environmental effects), albeit in a two-stage procedure in the genomic analyses (see later). The new pedigree-derived breeding values also adjust for these additional model terms, but in a single step model. Crucially, the two approaches (genomic and pedigree-derived EBVs) tell essentially the same story (Figure 2, Table S4, Table S5, Fig S1).

L125-126: I think this still has to be shown.

We can see that the reviewer is unconvinced by this point, and so we have removed the sentence. Now that we have EBVs from the pedigree and genomic data, it does appear that the latter give more conservative results though, even when adopting best practice for the pedigree-derived data (e.g. Tables S4, S5, S7).

L170: If I understood correctly, the trait analysed here is under selection. Would one hence not expect that causal loci could deviate from HW-equilibrium? Why not include these?

*This is an interesting point. We only removed SNPs that were **very** out of HWE. Deviations this extreme are nearly always caused by poor clustering of the SNP genotypes which the SNP-calling software then struggles to accurately assign reliable genotypes with. For a SNP to be this far removed from HWE due to selection, it would have to be under exceptionally strong selection. The accuracy of the GEBVs is high and consistent with theoretical expectations (see uploaded companion manuscript, Figs 1a, 3), and they are highly correlated with pedigree EBVs. Thus it seems unlikely that the SNPs that were out of HWE has unaccounted for large effects on weight (which would presumably be the ones under strongest selection). The inclusion of SNPs very out of HWE could lead to biases that could cause false temporary trends. e.g. if the cause of the departure from HWE was poor clustering/genotype miscalling, and older samples were of lower quality and were less likely to be correctly scored, spurious associations between GEBVs and year could emerge. On balance, we think our approach is conservative.*

L181: Since the aim of this manuscript is to compare pedigree-based with genomic approaches, it would be instructive to report (or calculate) the accuracy of pedigree-based EBVs for comparison, see e.g. Gienapp et al. (2019).

Both reviewers suggested we do more comparisons with pedigree-based approaches, and we now do that by using MCMCglmm to estimate EBVs from the pedigree. The focus of this paper is more on detection of microevolutionary trends rather than the accuracy of EBVs, but the accompanying manuscript describes the accuracy of the GEBVs in far more detail; indeed that is the main focus of the other paper. We simply refer to the accuracy of the GEBVs on this line and reference the other paper, because it is important to reassure the reader that we have previously validated our approach.

L182-185: Why these choices for effect sizes and number of pseudo-observations? I am aware of analysis where effects sizes of SNPs could be either zero or non-zero but I found these four values a bit arbitrary. Why suddenly five pseudo-observations for the largest effect size?

The other reviewer also raised a question about the choice of priors. The key point is that the BayesR estimates of GEBVs are remarkably insensitive to the choice of priors. For example, the correlation in GEBVs between the parameters we used and the default parameters is 0.9968! (see Table S4 of the companion manuscript). We can see why the choices might seem arbitrary, but the default value for BayesR is a mixture of 4 distributions (our results are essentially identical if we use 3 or 5 distributions). We have found that choosing a value of five pseudo-observations in the largest

effect size enables better mixing of the MCMC chains when SNPs of medium-large effect are present (e.g. for horn length, Table S4 of companion manuscript). See also our response to the similar comment raised by the other reviewer.

L192-195: You here use BLUPs as phenotypes. I find this a bit surprising given the known issues with BLUPs. You discuss these related to EBVs but these limitations generally apply all BLUPs. It would be preferable to include fixed effects and random effects in the genomic prediction model. Alternatively, you could account for the prediction error in the BLUPs by using a Bayesian approach as you did when testing for temporal trends in GEBVs.

In an ideal world we would have been able to implement BayesR in a single model that included the non-genetic fixed and random effects, but that is not currently possible, which is why we ran BayesR on the adjusted phenotypes. The BGLR package does allow BayesB (the next best performing model in our companion paper), and other less-well performing genomic prediction models to be run with other fixed and random effects, but even then dealing with repeated measures is not straightforward. It is possible to 'trick' BGLR by creating an additional row in the genotype matrix for every repeated phenotypic measurement an individual has. When we explored that in the past, we actually found that the BGLR BayesB GEBVs were more weakly correlated with adult weight than the BayesR GEBVs used in this paper were, implying that the single step approach implemented in BGLR was less accurate than the two-step approach we ran with BayesR.

Accounting for the prediction error in the BLUPs would be very hard, because we would need to run a separate BayesR analysis (each of which has a long run time) multiple times (e.g. 1000 times), with each run using a different sample from the distribution of estimated BLUPs. This would be computationally intractable.

Most crucially to our overall findings though, our new analyses include pedigree-derived EBVs from MCMCglmm runs that do include the other fixed and random effects in the same model. These analyses support the temporal trends revealed by the GEBVs. They show that EBVs for adult weight have significantly changed, probably more than can be explained by drift. Thus they support the conclusions from the BayesR GEBVs although the latter appear to be a little more conservative.

L258-261: Are really these estimates reported in the tables? If the trend in the real data set was about 0.01 and the difference between real data and 'gene dropped data' was also around 0.01, then the expected trend under drift would be around zero. In the original re-analysis (Hadfield et al. 2010) and similar papers the 'drift trend' was reported directly. So, I now wonder whether the expected change in breeding values after correcting for drift would be essentially zero!

We're not quite sure what the reviewer means by "the 'drift trend' reported directly", although we think that this refers to the P values describing whether the genetic trends seen in Soay sheep are actually negative ($P = 0.283$; $P = 0.357$ accounting for drift). In our paper we report the same statistics in the columns P_{MCMC} and P_{drift} in Tables S4 and S5. We didn't initially report the difference between the observed slopes and the 'drift' slopes in the tables, although they were plotted in Fig 1c and S1 (now Fig2, S1). We now include the estimates of the difference between the observed slopes and the drift slope in Table S4 (see the column Δ). The drift slopes are, as expected by the referee, around 0, which means the difference between the observed and drift slope is around 0.010 (i.e. almost identical to the posterior mode of the observed slopes). Thus, the evidence for microevolutionary change due to selection is now much stronger than it was at the time of the Hadfield 2010 paper.

L290-291: I do not agree here. First, no formal statistical significance is reached. Second, see the comment just above.

We can see why the reviewer felt that we were overstating the evidence for a response greater than expected by drift. The new analyses using pedigree data do reach statistical significance, and thus we feel that the statement is now more justifiable. Note that even now we are quite cautious – "Thus while genetic drift cannot be ruled out..."

L300-302: This point has been made before, see e.g. Moore & Kukuk (2002) or Gienapp et al. (2017). BTW, I found your manuscript generally lacking some citations to earlier relevant studies of genomic prediction or 'genomic quantitative genetics' in wild populations.

We now cite both of these reviews. In addition, the introduction now cites some of the earlier genomic prediction studies in the wild (Bosse et al., 2017; Gienapp et al., 2019), as well as the review on the subject that has appeared in this special issue of Proceedings B (McGaugh et al. 2021).

L311-314: This actually contradicts your statement in the Introduction on L123-125. I am not familiar with all software for genomic prediction but this is definitely possible when fitting a GRM in e.g. ASReml or MCMCgImm, i.e. following a so-called GBLUP approach.

We have reworded this section. Of course, GBLUP models fitting a GRM can be run in the packages mentioned, but we were specifically meaning the type of model where individual SNP effects are estimated from a training population and then used to predict GEBVs in a test population. Such approaches avoid the assumption of a pseudo-infinitesimal model of trait architecture and in our system have greater accuracy, especially when major loci are segregating (see accompanying manuscript). We should have been clearer on this point.

L316-317: See above comment about using BLUPs from a Bayesian model.

As described above, we agree that the uncertainty in the BLUPs that were fed into BayesR were not accounted for, although hopefully the uncertainty in the subsequent GEBVs of those BLUPs are. The pedigree-based analyses are a single-step model with all random and fixed effects fitted and so they do account for all of the uncertainty. Note that they support the genomic analyses i.e. they support the evidence that breeding values for weight have increased, so we can be confident that the reported microevolutionary trends were not caused by the way the phenotypes were pre-adjusted prior to the BayesR runs.

L358: I am not really convinced by this evidence...

Now that the pedigree-based analyses support this evidence (compared to the genomic estimates of breeding values, they actually predict a larger response, and are more statistically significant) we hope that the reviewer is more convinced.

cited literature

Moore AJ, Kulkarni PF (2002) Quantitative genetic analysis of natural populations. *Nature Rev. Genet.* 3, 971–978 (2002).

P Gienapp, S Fior, F Guillaume, JR Lasky, VL Sork, K Csilléry (2017) Genomic quantitative genetics to study evolution in the wild. *Trends in Ecology and Evolution* 32: 897-908

P Gienapp, MPL Calus, VN Laine, ME Visser (2019) Genomic selection on breeding time in a wild bird population. *Evolution Letters* 3: 142-151

Referee: 2

Comments to the Author(s)

In their study "Using genomic prediction to detect microevolutionary change of a quantitative trait" Hunter and colleagues investigate the microevolutionary responses to natural selection through genomic estimated breeding values for adult weight in a 35-year dataset of Soay sheep. While the empirical data showed decreased body size, genome derived breeding values suggest an increase in the trait. The authors further use gene dropping simulations to test whether the estimated change was the result of genetic drift. The authors conclude that the increase in genetic breeding values was larger than expected by drift and attribute this to cryptic microevolution. The overall idea of the study is interesting and addresses important questions in evolutionary genetics and the

introduction, methods and discussion section are well written. However, the extremely short results section does not include sufficient information to assess the results of the different analyses performed. In particular the discrepancy between observed phenotypic change and predicted change has to be thoroughly evaluated to come to the "cryptic microevolution" interpretation and exclude methodological artefacts as reason for opposing directions. A more in-depth analysis of the accuracy of genomic breeding values, comparison with previously estimated pedigree based breeding values and more information about the simulation results should be added.

The reviewer has some overlapping comments with Reviewer 1, especially with regards to more details being needed about the comparison between genomic and pedigree EBVs. We now report complementary results from pedigree-derived EBVs which support the results from the genomic data. The reviewer asked for more details of the accuracy of our genomic EBVs, and we have now uploaded a complementary manuscript, provisionally accepted at Molecular Ecology, which describes an analysis of genomic prediction for weight and 7 other traits, comparing 5 different genomic prediction methods. We have extended the Results section by about 50%, and the Figures have been extended to include plots of the trends derived from the genomic and pedigree EBVs.

Major comments:

- The study is highly dependent on accurate estimation of genomic estimated breeding values (gEBVs), however the accuracy of the estimates was not assessed in the current manuscript. It would be very useful to see prediction accuracies based on cross validation results.

We should have made it clearer that we have addressed this in a different manuscript, where we evaluated the accuracy of genomic prediction of 8 traits (with a range of genetic architectures) including weight, the focal trait in this study. On lines 164-166, we briefly explained how we assessed the accuracy of our genomic prediction and we referenced a preprint we had uploaded to bioRxiv. That preprint has now been provisionally accepted at Molecular Ecology, and a revised manuscript will be submitted in the first half of October 2021. Clearly it would have been a good idea if we had uploaded that manuscript as part of this submission as well though, rather than just reference the preprint. Briefly, we compared 5 different methods of genomic prediction, showed that BayesR was the most accurate of the methods we used, and show that all traits have accuracies in line with (or perhaps slightly better than) theoretical explanations. The accuracy of our GEBVs for weight (~0.64) is generally at least as good as is seen in domestic sheep populations e.g. (Moghaddar et al., 2017; Riggio et al., 2014)

We don't feel that this manuscript is the right place to present the cross-validation results in any detail, as that is the focus of the other paper. In addition, the study is now less dependent on the accuracy of the GEBVs, as the main conclusions are supported by addition of the analyses using the pedigree EBVs.

- The methods describe different distributions for effect sizes in the gEBV estimation. how did these affect the outcome?

Reviewer 1 asked a similar question. BayesR assumes that the SNP effects come from >1 distribution, including some SNPs having an effect size of zero. The default in the software is to assume 4 distributions. We ran additional models with 3 and with 5 distributions, and the GEBVs are not at all sensitive to these changes.

The correlation between GEBVs

determined from three and from four distributions was 0.9965; correlation between GEBVs determined from three and from five distributions was 0.9935 and the correlation between GEBVs estimated from four and from five distributions was 0.9897. Clearly the number of distributions does not make a great difference to the final GEBVs that are obtained.

- I agree that eEBVs provide a promising path to study evolutionary trends, but given the statement that this study "introduced a genomic approach to studying microevolutionary trends", I would have expected a more detailed evaluation of the factors that influence the accuracy of the

results. Probably including simulations evaluating the robustness. I think the authors have a great dataset on hand to address these questions.

Factors that influence the accuracy are covered in the other paper, but they include the heritability of the trait, the size of the training population (this is especially important), the architecture of the trait (major alleles or polygenic), the marker density and the method used for predicting the GEBVs.

- How many effect sizes were used/estimated? Can you maybe elaborate on the n (individuals) $<$ p (markers) problem for effect size estimation and how that might influence your allele dropping results?

BayesR reports effect sizes for all of the SNPs; these are estimated simultaneously, rather than one at a time. For each SNP, a distribution of possible effect sizes is obtained, and so for each simulation a different sample from the distribution of effect sizes is used. All of the SNPs are used in the simulations, because haplotypes are dropped down the pedigree. It appears that the drift simulations using the estimated SNP effects are slightly more conservative (see Table S7) than the traditional method (as in Hadfield 2010) where breeding values are assigned to founders and their progeny are given midparental values.

- It is not clear to me what the "test set" was used for in this study. This set cannot be used to assess the accuracy of the predictions.

The test set was the set of individuals that were genotyped but did not have recorded phenotypes (see last paragraph of Methods section 'Genomic Prediction of weight GEBVs'. We did not use them to assess accuracy in this study, and of course the reviewer is correct to say that without phenotypes they cannot be used to assess accuracy. In the other paper, we used just the genotyped and phenotyped individuals, then treated a fraction of them as if they were not phenotyped, and performed cross-validation by looking at the correlation between their predicted breeding values and their true phenotypes; i.e. we used the standard method for assessing accuracy of genomic prediction.

We use the terms 'training' and 'test' to be consistent with most of the literature on genomic prediction, but given that this paper does not test the accuracy, we could use different terms such as 'genotyped only' and 'genotyped and phenotyped' to replace 'test' and 'training' in our paper.

Minor comments:

- The title suggests that "genomic prediction" has been used, but I think this is technically not the case as not the phenotype prediction was used but rather the effect size estimation.

To clarify, we did use genomic prediction of the estimated breeding values underlying the phenotype. Of course, these breeding values were derived from the estimated SNP effects (as is true for many genomic prediction methods – BayesA, BayesB, BayesR, etc), but it is certainly the breeding values that are estimated and then used to explore the evolutionary genetic trends of weight.

- Please clearly state what data has been published before and what is unpublished. do I understand correctly that genotyping data for the full set is so far unpublished, or has it been used in other studies before?

The full genotyping dataset has been published previously, both in the manuscript accompanying this one, but also in numerous GWAS and molecular quantitative genetic studies of Soay sheep (Bérénos et al., 2015; Bérénos et al., 2014; Johnston et al., 2016; Johnston et al., 2013; Sparks et al., 2019). We have edited the text to make it clear that the cited papers include links to publically available genotype data.

- I. 323ff this should probably go into the results

We included the description of the rate of evolutionary change in Haldanes in the Discussion, as we wanted to put it in context of other studies, but we can see why the reviewer thinks the initial part, stating the rate of change, should be in the Results section. We do feel that a 1-2 line description of the rate in the Results and a subsequent 1-2 line discussion of relating that rate to other studies, appearing later in the paper, would break up the flow of the point being made though. We would prefer to keep the text as it is, but if necessary will move the first part to the Results.

- Supplementary figures and tables are not referenced in the text.

Good point. We now point the reader to the relevant Tables and Figures in the supplementary material, rather than just refer them to the SM more generally.

- is the correlation line in fig. 1 a based on year/means or individual values? Slope for females and males look much less severe when plotted from the supplied "Weight_Life.txt" data.

Figure 1a shows the cohort means and SEs and the line is the regression of cohort means on year. Regardless of whether the analysis is conducted on cohort means or individual values, the slopes are very similar (-0.075 kg per year; SE 0.0185 for cohort means; -0.067 kg per year; SE 0.035 for individual values). We felt plotting cohort means was neater than plotting every individual, and also provided consistency with previous papers, both in this system (Wilson et al., 2007) and other similar papers exploring microevolutionary trends in EBVs e.g. (Pigeon et al., 2016).

- in the supplementary file "Weight_Life.txt" non of the individuals has both a weight value and weight_gebv. How come? The correlation between those 2 would be interesting.

The readme.txt file explains the column headings. weight_gebv is the genomic breeding values of just the 'test' population individuals, ie. Those without a phenotype. Thus, it is not possible to calculate the correlation between that variable and actual weights. The weight column is the adjusted weight for individuals who did have a known weight i.e. the training population. More generally though, the correlation between genomic breeding values and known phenotypes was explored in a lot of detail in the other manuscript (uploaded with this submission), as this was the approach used for cross-validation estimates of genomic prediction accuracy.

References

- Abecasis, G. R., Cherny, S. S., Cookson, W. O., & Cardon, L. R. (2002). Merlin--rapid analysis of dense genetic maps using sparse gene flow trees. *Nat Genet*, *30*(1), 97-101. doi:10.1038/ng786
- Bérénois, C., Ellis, P. A., Pilkington, J. G., Lee, S. H., Gratten, J., & Pemberton, J. M. (2015). Heterogeneity of genetic architecture of body size traits in a free-living population. *Molecular Ecology*, *24*(8), 1810-1830. doi:10.1111/mec.13146
- Bérénois, C., Ellis, P. A., Pilkington, J. G., & Pemberton, J. M. (2014). Estimating quantitative genetic parameters in wild populations: a comparison of pedigree and genomic approaches. *Molecular Ecology*, *23*(14), 3434-3451. doi:10.1111/mec.12827
- Bosse, M., Spurgin, L. G., Laine, V. N., Cole, E. F., Firth, J. A., Gienapp, P., . . . Slate, J. (2017). Recent natural selection causes adaptive evolution of an avian polygenic trait. *Science*, *358*(6361), 365--368. doi:10.1126/science.aal3298

- Gienapp, P., Calus, M. P. L., Laine, V. N., & Visser, M. E. (2019). Genomic selection on breeding time in a wild bird population. *Evol Lett*, *3*(2), 142-151. doi:10.1002/evl3.103
- Hadfield, J. D. (2010). MCMC Methods for Multi-Response Generalized Linear Mixed Models: The MCMCglmm R Package. *Journal of Statistical Software*, *33*(2), 1-22. doi:10.18637/jss.v033.i02
- Hadfield, J. D., Wilson, A. J., Garant, D., Sheldon, B. C., & Kruuk, L. E. B. (2010). The Misuse of BLUP in Ecology and Evolution. *American Naturalist*, *175*(1), 116-125. doi:10.1086/648604
- Johnston, S. E., Bérénos, C., Slate, J., & Pemberton, J. M. (2016). Conserved Genetic Architecture Underlying Individual Recombination Rate Variation in a Wild Population of Soay Sheep (*Ovis aries*). *Genetics*, *203*(1), 583-598. doi:10.1534/genetics.115.185553
- Johnston, S. E., Gratten, J., Bérénos, C., Pilkington, J. G., Clutton-Brock, T. H., Pemberton, J. M., & Slate, J. (2013). Life history trade-offs at a single locus maintain sexually selected genetic variation. *Nature*, *502*(7469), 93-95. doi:10.1038/nature12489
- MacCluer, J. W., VandeBerg, J. L., Read, B., & Ryder, O. A. (1986). Pedigree analysis by computer simulation. *Zoo Biology*, *5*(2), 147-160. doi:<https://doi.org/10.1002/zoo.1430050209>
- McGaugh, S. E., Lorenz, A. J., & Flagel, L. E. (2021). The utility of genomic prediction models in evolutionary genetics. *Proc Biol Sci*, *288*(1956), 20210693. doi:10.1098/rspb.2021.0693
- Moghaddar, N., Swan, A. A., & van der Werf, J. H. J. (2017). Genomic prediction from observed and imputed high-density ovine genotypes. *Genet Sel Evol*, *49*(1), 40. doi:10.1186/s12711-017-0315-4
- Ozgul, A., Tuljapurkar, S., Benton, T. G., Pemberton, J. M., Clutton-Brock, T. H., & Coulson, T. (2009). The Dynamics of Phenotypic Change and the Shrinking Sheep of St. Kilda. *Science*, *325*(5939), 464-467. doi:10.1126/science.1173668
- Pigeon, G., Festa-Bianchet, M., Coltman, D. W., & Pelletier, F. (2016). Intense selective hunting leads to artificial evolution in horn size. *Evolutionary Applications*, *9*(4), 521-530. doi:10.1111/eva.12358
- Postma, E. (2006). Implications of the difference between true and predicted breeding values for the study of natural selection and micro-evolution. *Journal of Evolutionary Biology*, *19*(2), 309-320. doi:10.1111/j.1420-9101.2005.01007.x
- Riggio, V., Abdel-Aziz, M., Matika, O., Moreno, C. R., Carta, A., & Bishop, S. C. (2014). Accuracy of genomic prediction within and across populations for nematode resistance and body weight traits in sheep. *Animal*, *8*(4), 520-528. doi:10.1017/S1751731114000081
- Sparks, A. M., Watt, K., Sinclair, R., Pilkington, J. G., Pemberton, J. M., McNeilly, T. N., . . . Johnston, S. E. (2019). The genetic architecture of helminth-specific immune responses in a wild population of Soay sheep (*Ovis aries*). *PLoS Genet*, *15*(11), e1008461. doi:10.1371/journal.pgen.1008461

Wilson, A. J., Pemberton, J. M., Pilkington, J. G., Clutton-Brock, T. H., Coltman, D. W., & Kruuk, L. E. B. (2007). Quantitative genetics of growth and cryptic evolution of body size in an island population. *Evolutionary Ecology*, *21*(3), 337-356. doi:10.1007/s10682-006-9106-z

The
University
Of
Sheffield.

School of Biosciences.

Prof Gary Carvalho
c/o Shalene Singh-Shepherd
Editorial Office
Proceedings B

Professor Jon Slate
School of Biosciences
Alfred Denny Building
University of Sheffield
Western Bank
Sheffield
S10 2TN

Telephone: +44 (0) 114 222 0048

Email: j.slate@sheffield.ac.uk

<https://www.sheffield.ac.uk/biosciences>

12 January 2022

Manuscript Number: Proceedings B RSPB-2021-2201

Dear Prof Carvalho / Senior Editor Singh-Shepherd

Thank you for giving us the opportunity to submit a further revision of our manuscript *Using genomic prediction to detect microevolutionary change of a quantitative trait*. The Associate Editor's and Reviewer's comments were most constructive, and we are confident we have further improved the manuscript as a result of their suggestions. Below we respond to each of the Associate Editor's and reviewers' points - our responses are in blue, italicised font. We have also submitted a clean and marked-up version of the manuscript and a revised supplementary information document. We have again undertaken some new analyses while revising the manuscript and have therefore updated the files and code on the Dryad submission linked to this manuscript. We very much hope that this revision has addressed all of the outstanding points from before.

Your sincerely

Jon Slate (on behalf of all of the authors)

AE and Reviewers' Comments and Our Responses

Associate Editor Board Member

Comments to Author:

Overall the authors have done a great job incorporating suggestions from reviewers. However, the two reviewers raise a number of points that are important to address. The main remaining concern is that Reviewer 1 would like to see a reanalysis of the same time period considered

in Wilson et al. 2007 and Hadfield et al 2010 (1985-2005). I recognize that addressing this comment may require substantial work; however I agree that the framing of the manuscript sets up the expectation that the same dataset will be reanalyzed. I think that directly comparing results to previous papers and then discussing the results after expanding the dataset would make the manuscript stronger.

We are pleased that it was felt we had done a great job incorporating the previous suggestions. Hopefully, we have done likewise with the more recent ones. As suggested above, we have now included an analysis that includes the 1985-2005 period. The main conclusion from that new analysis is that the rates of evolutionary change are actually very similar for the 'Wilson Years' (1985-2005) [1], the period we analysed in the original manuscript (1990-2015) and the 'Post-Wilson' period (2005-2015). There are differences in statistical significance but this reflects differences in sample sizes more than differences in the magnitude of effects. We have added a section to the Discussion where we consider why it is that Hadfield's 2010 analysis of the 1985-2005 period could not reject stasis [2] , but our analysis of 1985-2005 probably does reject stasis in favour of a microevolutionary change that might be stronger than expected by drift. It is important to note that regardless of the time period analysed or the approach used to estimate breeding values, there is a consistent pattern of increasing breeding values for adult weight.

I have a few additional comments:

For the tests of microevolutionary change, since pedigree-derived EBVs are simulated by assigning progeny midparental values, I wonder if it would be useful to use the same approach for the GEBV simulations and compare those results to the gene dropping results in the main text instead of the supplement. That way the main text could more directly compare tests of microevolutionary change using pedigree-derived EBVs vs GEBVs, and then compare tests using different ways of simulating expectations for GEBVs. The observed differences between pedigree and genomic EBVs may be due to both differences in the EBV estimates and the method used for simulating change under drift, and it would be nice to be able to disentangle the two.

Yes, we had wondered about doing this. Initially we were concerned that it might be confusing to compare genomic-derived EBVs to patterns simulated from pedigree derived midparental values and vice-versa (pedigree-derived EBVs to patterns derived from simulations that used effects of individual loci). However, it does make sense to independently consider the two factors at play here – the method used to estimate EBVs and the method used to simulate patterns caused by drift alone. We now use the notation $P_{drift_genomic}$ and $P_{drift_pedigree}$ to distinguish between tests that use the two alternative ways of estimating drift-determined trends.

For the discussion of pedigree depth, it would be helpful to add a figure showing the distribution of phenotyped close relatives for individuals of different birth cohorts.

This was an excellent suggestion, and we have now included a plot that does this (supplementary information Fig 5). For each cohort we have plotted the distribution of the number of phenotyped relatives with $r \geq 0.125$ (equivalent to cousins). It is noticeable that the last three years of data (2013-2015) show a much lower mean number of phenotyped close relatives, which is consistent with the idea that pedigree EBVs in these cohorts are closer to

the overall mean because there are less data available with which to accurately estimate breeding values.

If the primary aim of the paper is to introduce a genomic approach to studying microevolutionary change, it would be nice to include a paragraph in the discussion about requirements and caveats of such an approach (e.g., a discussion of minimum marker numbers and sample sizes, especially as it is difficult to accurately estimate small effect sizes with the sample sizes used in most studies of natural populations).

We are reluctant to provide specific numbers, because the minimum number of markers and individuals will vary enormously between study systems, such that any suggested numbers we offer run the risk of being unsuitable for the majority of other studies. Obviously the effect size (in terms of the amount of evolutionary change) will be critically important, but so too will be the ability to reliably estimate GEBVs and that is dependent on, among other things, the sample size, the trait's heritability, the amount of linkage disequilibrium in the genome, and the distribution of effect sizes of the (unknown) loci that affect the trait. However, in light of our re-analysis (see below; Reviewer 3's comments) of the magnitude of our microevolutionary trend, relative to others, we are now more cautious in our advocacy of the approach. When measured in Haldanes, the genetic increase in adult weight is in the top ~11% of microevolutionary responses to selection described in the literature, albeit it with the caveat that most of the studies we compare our estimate to are based on phenotypic rather than genetic change. Furthermore, we have already shown that the accuracy of our GEBVs is relatively high compared to e.g. domestic livestock populations [3]. Thus our system is probably one in which microevolutionary trends are relatively easy to detect.

Lines 262-263: It is unclear why phased haplotypes were randomly assigned to founders in the gene dropping analyses; why not use the actual phased haplotypes for each founder?

We could have done this for founders whose identity was known and who were genotyped, but sometimes founders are unknown (i.e. dummy IDs are assigned when paternity or maternity of the lamb is undetermined) and also some founders may not have been genotyped on the SNP chip (Table S1 shows how relatively few individuals in cohorts before 1990 were genotyped).

Line 273: Why between 2 and 10 years? Is this the typical breeding lifespan of Soay sheep?

Yes – lambs and yearlings sometimes achieve reproductive success, but this is relatively rare. Few animals live to the age of 10.

Reviewer(s)' Comments to Author:

Referee: 3

Comments to the Author(s).

This ms explores the potential for using genomic prediction to detect microevolutionary trends using a long term study of Soay sheep.

I was not one of the original reviewers on this so have only looked at the revised ms and the authors response letters.

In my view the authors have done a great job in addressing all the comments from the reviewers in a very comprehensive manner and I think the ms is much improved as a result.

We thank this new reviewer for being so positive about the revision, and in particular for taking the time to look at our response and revisions to the reviews of the original version.

I just have a few small things:

Regarding the Haldane estimate provided in the discussion I am not sure this is correctly derived? In Gingerich 1993 (that the authors cite) the log difference in mean is used, whereas here it is the difference in average (GEBV) between generations here (slope):

rate(h) = $(\ln x_2 / \text{slnx}) - (\ln x_1 / \text{slnx}) / t_2 - t_1$, where $\ln x_2$ is the natural log of the phenotype at time point 2 and slnx is the pooled SD of $\ln x_1$ s and $\ln x_2$ s.

The authors correctly take generation time into account to obtain a measure of change per generation but I think that this is not comparable to other estimates from the review article by Hendry et al. if they (presumably) used the log difference? Maybe worth to check up.

Also, Using the approach as authors on the phenotypic trend instead I get a Haldane estimate of $0.067/7.4*4 = 0.036$, which is pretty high.

My point is that I think the argument of genomic prediction being useful even at low intensities of selection does not necessarily follow from this calculation.

Thank you for pointing out that a (natural) log scale should perhaps be used here and for encouraging us to think more carefully about the rates of evolutionary change.

We actually hadn't cited the Gingerich (1993) paper, and the paper containing the data that we did compare our estimates to (Hendry et al. 2008) [4] does not make it clear whether the phenotypes they studied were log-transformed. However, it seems prudent to do so. Using log-transformed data, we agree that the rate of evolutionary change was greater than we had appreciated.

- 1) *We start by estimating the mean August weight (25.57 kg) in our data set and adding the genomic breeding values of each individual to this overall mean.*
- 2) *Next we natural-log transformed the mean weight + GEBVs described in step 1. These values (of log-transformed GEBVs) had a mean 3.205 and SD of 0.034.*
- 3) *We then performed a linear regression of log-transformed GEBVs against year (mean-centred), and obtained a regression coefficient of 0.00042.*
- 4) *Dividing the regression coefficient by the SD of log-transformed GEBVs and multiplying this by 4 (the mean generation time) gives $0.00042 / 0.034 * 4 = 0.049$. This is the equivalent of*

$$h = \frac{\frac{\ln(x_2)}{SD \ln(x)} - \frac{\ln(x_1)}{SD \ln(x)}}{t_2 - t_1}$$

where x_2 is the mean log-transformed weight in 1976, x_1 is the mean log-transformed weight in 2015 and $SD \ln(x)$ is the standard deviation of log-transformed weight in the entire dataset. $t_2 - t_1$ is the number of years in the time series (39 years), divided by the mean generation time (4 years).

The paper by Hendry and colleagues, reporting studies of microevolutionary change in wild populations has recently been updated by Sanderson and colleagues to include more studies [5]. The rate of change we observe, of $h = 0.049$ Haldanes, is exceeded by only around 11% of

the estimates in the paper by Sanderson and colleagues. Thus, it now seems reasonable to point out, as suggested by the reviewer, that the trend we are working with is relatively large, and it may be harder to detect significant changes in other datasets where the response to selection is weaker.

The explanation for how we've reached this conclusion is obviously now quite long, so we have included a slightly edited version of the information above as part of the supplementary material, rather than report it all in the main text.

Throughout: perhaps it makes more sense to convert estimates of changes into grams instead of e.g. 0.01kg per year even if the slope of course is estimating kg/year. Given weight of the sheep we really are talking minuscule modest changes.

We did consider this from the outset, but felt that it made sense to stick to Kg given that (i) this is the SI unit of mass and (ii) We measure, report and describe sheep weights in Kg rather than g. (iii) Furthermore, using Kg makes our paper consistent with previous ones studying the same trait.

line 51: then not than

Thank you – we've changed this.

line 53: I would cite Henderson's original work as well, it was rediscovered by the wild quantitative group.

Good point – we now cite Henderson (1975) [6] as well.

line 207: I understood from the response to reviewers that it is very difficult to include uncertainty in the BLUPs into account with the current software available. Nonetheless, it is striking given what we know how ignoring uncertainty in EBVs caused a lot of wrong inferences to be made that, we are still in that situation with GEBVs. I agree with the authors though that the added EBV analysis from MCMCGLMM go some way to support the inferences from GEBVs that ignore the uncertainty in BLUPs. One can only hope this is not specific to the situation here but applies more generally...perhaps a statement to this effect can be added to where this is discussed (line 369-366).

We have added the statement: "The pedigree-derived EBVs (which are from a single-stage model) and the GEBVs (which are not) produced similar conclusions. In fact, our conclusions from the GEBVs seem to be more conservative than those from the pedigree EBVs, but of course we cannot be certain that would be the case in other systems".

line 214-215: maybe I am missing something but in the 'phenotype' for the GEBVs you include birth year and capture year as random effect whereas for the EBV analysis you use birth year as random effect but capture year as fixed effect? Why the difference?

There is perhaps a misunderstanding here. In the models where we estimated both GEBVs and pedigree EBVs we included capture year and birth year as random effects. Capture age was fitted as a fixed effect. We think perhaps the reviewer has read capture age as capture year. The main point though, is that the models for GEBVs and pedigree EBVs were consistent.

line 228-229: so effectively a one-tailed test?

Yes, the test is one-tailed. It was also one-tailed in the Hadfield (2010) paper that we cite. Two-tailed tests would not have a big effect on our conclusions. The trends in GEBVs/EBVs over the

duration of the study would still be significant and the evidence that the trends were greater than might be expected due to genetic drift would be reasonably strong but not conclusive.

Line 348: but reconstructing a pedigree once you have all the molecular marker data is not really very difficult or time consuming so I don't really understand this first argument. I would say the potential to avoid biases is of much greater importance no?

Actually, we think the first argument is an important one. In many systems, there is a low chance that a pedigree can be recovered, because parents may not have been sampled. e.g. obligate spawners, outbreeding plants, birds with high dispersal rates. Thus, having molecular marker data from phenotyped individuals means that we can perform genomic prediction and look for trends in GEBVs in systems where pedigree construction is likely to be either impossible or very incomplete. The point we wanted to make, and now try to emphasise more strongly, is that this approach should enable us to broaden the types of system where analyses of microevolutionary trends can be performed. A very high proportion of studies to date have been conducted in ungulates and birds.

line 397: what is a "largely polygenic trait"?

The point we were trying to articulate is that some genomic regions explaining significant additive variation have been found ([3, 7], which perhaps explain a little more variation than expected compared to a trait where each chromosome's contribution was proportional to its size. Nonetheless, most chromosomes contribute some (measurable) variation, and the trait is certainly not one where a handful of loci explain most of the variation. Perhaps it would be simpler if we removed the word 'largely' to avoid any confusion.

Line 414: so, 10g per year?

Yes

Referee: 1

Comments to the Author(s).

Dear Authors,

My main concern with the previous version of your manuscript was the lacking comparison between pedigree-based and genomic quantitative genetic analyses. In this respect your manuscript has been considerably improved as both analyses are presented. As I wrote earlier ignoring new data and excluding them from the analysis is generally not a good idea. However, since your manuscript heavily refers back to the analysis of Wilson et al. (2007) it would be desirable if the same period than in this paper (1985-2005) would have been analysed with the genomic quantitative genetic analysis. Now, you present a pedigree-based analysis of a different period including the recent years (with the reversed phenotypic trend). While in the original analysis the genetic trend was not significant (after properly taking drift into account), it is now significant in the pedigree-based analysis. It would make a far better point for the usefulness of genomic quantitative genetic approaches if the genomic approach could demonstrate a genetic trend while the pedigree-based approach could not. Actually, now the pedigree-based trend is significant while the genomic trend is not (although this may be a result of the more conservative gene-dropping test).

We are pleased that the reviewer feels the manuscript has been improved. We now present the data from 1985-2005 as well, to make a fair comparison with the Wilson et al. study [1]. The data from 1985-2005 are broadly consistent with all of the other time periods reported in the earlier manuscript – a significant increase in EBVs (genomic or pedigree-derived) that is on the margins of what might be expected under genetic drift alone. The actual magnitude of change in EBVs is slightly greater during this period than the other periods analysed (1980-2015, 1990-2015 and 2005-2015), but essentially the different periods tell the same story.

While we can understand the reviewer's argument that it would be 'better' for the usefulness of genomic prediction if it showed a significant trend when the pedigree-based approach did not, we actually find it reassuring that the two approaches tell consistent stories. There will be situations where only a genomic approach can be used (see the response to Reviewer 3 about systems where pedigree construction is impossible), so we see it as a good sign that the two approaches provide similar answers to the question of whether microevolutionary change has occurred. Where the approaches do yield different results we have found a possible explanation that is technical rather than biological.

My other comments concerning the analyses (e.g. choice of prior, two-step analysis) have been satisfyingly addressed. However a few specific comments arose again:

We are pleased that the other comments have been addressed. Below we address the remaining/new comments.

L97 Ref 21 seems to be wrong in the list of references. Please double-check also all other references as I did not check all (I find numbered references generally cumbersome...).

Thank you – we had cited the wrong reference here (Browning and Browning) and have replaced it with the correct one (Wilson et al. 2007).

L100-104 I am puzzled by this added explanation that refers to the recently reversed phenotypic trend, while this paragraph deals with the original paper by Wilson and later the re-analysis by Hadfield. To my understanding the contrasted phenotypic and genetic trends were thought to be caused by deteriorating environmental conditions. Should this addition not better be moved to L389-391 where you discuss this? The explanation given there differs from what you write here, though.

Actually the Wilson paper was cautious about stating causes of the phenotypic decline in body size and did not invoke environmental deterioration as the explanation. The paper we cite, by Ozgul and colleagues [8], suggested that the cause may be a complex interaction between weather, density and a change in the demographic profile of the population. In recent years, an improved environment may mean that lighter sheep that once would have died as young animals are more likely to survive to adulthood. Thus, the mean weight is lower than if only the largest animals survived to adulthood. It is this change in demography, rather than a deteriorating environment that seems to be the most likely explanation for the phenotypic trend. We would prefer to retain this information in the Introduction rather than move it to the Discussion, as our data do not really speak to the causes of non-genetic changes in the phenotype. However, we felt it important to explain the phenotypic trend in the Introduction. We don't feel that the information on the non-genetic causes of the phenotypic trends differs between the Introduction and Discussion. In both places we emphasise there are non-genetic drivers of the decline in the phenotype.

L415 'introduced' is a bit overstated as it implies that something conceptually novel was used.

As far as we are aware, this is the first paper to use GEBVs to explore temporal microevolutionary trends, but we take the point; GEBVs have been widely used in other contexts. We have replaced 'introduced' with 'demonstrated'.

L418/419 This evidence is partly only convincing because of the significant genetic trend from the pedigree-based analysis.

Both the pedigree and genomic approaches show statistically significant increases in adult body weight breeding values over the study period. However, the evidence that the trend is not caused by genetic drift is slightly stronger with the pedigree-based approaches. We don't think the text says anything inconsistent with the data.

References

- [1] Wilson, A.J., Pemberton, J.M., Pilkington, J.G., Clutton-Brock, T.H., Coltman, D.W. & Kruuk, L.E.B. 2007 Quantitative genetics of growth and cryptic evolution of body size in an island population. *Evol. Ecol.* **21**, 337-356. (doi:10.1007/s10682-006-9106-z).
- [2] Hadfield, J.D., Wilson, A.J., Garant, D., Sheldon, B.C. & Kruuk, L.E.B. 2010 The Misuse of BLUP in Ecology and Evolution. *Am. Nat.* **175**, 116-125. (doi:10.1086/648604).
- [3] Ashraf, B., Hunter, D.C., Berenos, C., Ellis, P.A., Johnston, S.E., Pilkington, J.G., Pemberton, J.M. & Slate, J. 2021 Genomic prediction in the wild: A case study in Soay sheep. *Mol Ecol.* (doi:10.1111/mec.16262).
- [4] Hendry, A.P., Farrugia, T.J. & Kinnison, M.T. 2008 Human influences on rates of phenotypic change in wild animal populations. *Mol Ecol* **17**, 20-29. (doi:10.1111/j.1365-294X.2007.03428.x).
- [5] Sanderson, S., Beausoleil, M.-O., O'Dea, R.E., Wood, Z.T., Correa, C., Frankel, V., Gorné, L.D., Haines, G.E., Kinnison, M.T., Oke, K.B., et al. 2022 The pace of modern life, revisited. *Molec. Ecol.* **31**, 1028-1043. (doi:<https://doi.org/10.1111/mec.16299>).
- [6] Henderson, C.R. 1975 Best linear unbiased estimation and prediction under a selection model. *Biometrics* **31**, 423-447. (doi:10.2307/2529430).
- [7] Béréanos, C., Ellis, P.A., Pilkington, J.G., Lee, S.H., Gratten, J. & Pemberton, J.M. 2015 Heterogeneity of genetic architecture of body size traits in a free-living population. *Molec. Ecol.* **24**, 1810-1830. (doi:10.1111/mec.13146).
- [8] Ozgul, A., Tuljapurkar, S., Benton, T.G., Pemberton, J.M., Clutton-Brock, T.H. & Coulson, T. 2009 The Dynamics of Phenotypic Change and the Shrinking Sheep of St. Kilda. *Science* **325**, 464-467. (doi:10.1126/science.1173668).

The
University
Of
Sheffield.

School of Biosciences.

Prof Gary Carvalho
c/o Shalene Singh-Shepherd
Editorial Office
Proceedings B

Professor Jon Slate
School of Biosciences
Alfred Denny Building
University of Sheffield
Western Bank
Sheffield
S10 2TN

Telephone: +44 (0) 114 222 0048

Email: j.slate@sheffield.ac.uk

<https://www.sheffield.ac.uk/biosciences>

29th March 2022

Manuscript Number: Proceedings B RSPB-2021-2201 / RSPB-2022-0330

Dear Prof Carvalho / Senior Editor Singh-Shepherd

Thank you for giving us the opportunity to submit a further revision of our manuscript *Using genomic prediction to detect microevolutionary change of a quantitative trait*. The Associate Editor and Reviewer 4 recommended some minor changes, which we hope we have now accommodated. Below we respond to each of the Associate Editor's and reviewers' points - our responses are in blue, italicised font. We have also submitted a clean and marked-up version of the manuscript and a revised supplementary information document. We very much hope that this revision has addressed all of the outstanding points and that the manuscript is now ready for acceptance.

Your sincerely

Jon Slate (on behalf of all of the authors)

Associate Editor Board Member

Comments to Author:

The authors have done a thorough job of addressing previous suggestions, and I think this manuscript will be a valuable contribution to our special feature. This manuscript has also been evaluated by an expert reviewer, who has two methodological concerns and a number of suggestions for additional clarification. Their detailed comments seem addressable, and I think it would be acceptable to not redo/add analyses and instead just add an acknowledgment of other methods for estimating GEBVs and a brief discussion of the robustness of the gene dropping simulations to errors in genetic architecture.

We thank the Associate Editor for their positive comments and suggestion that the reviewer's comments can be addressed without further analyses. Of course, we have endeavoured to edit the manuscript to address all of the comments made by the Associate Editor and Reviewer 4. Below we outline all of the changes we have made.

I have just a few small additional comments:

Lines 417-418: This sentence does not seem particularly helpful. While I agree with the authors that minimum sample sizes and marker numbers are highly dependent on the study system, I wonder if it would be more useful (for readers) to replace this sentence with a suggestion that folks run simulations of traits with different genetic architectures and varying levels of LD to help assess sample requirements.

We appreciate our original statement may have seemed unhelpful. We have added the following sentence:

'However, we do recommend that researchers first establish the accuracy of GEBVs by cross-validation, or attempt to estimate the likely accuracy of GEBVs using formulae that predict the likely accuracy under given genetic architectures, genome sizes and effective population sizes (Daetwyler et al., 2010).'

The reference is to a paper that provides useful formulae for estimating the likely accuracy of genomic prediction, provided the genome length and effective population size can be estimated (which of course, will usually be the case). The formula assumes a highly polygenic trait, but can easily be adapted for genetic architectures with far fewer QTL.

Tables S4 & S5: Might be good to use the same terminology as the main text ($P_{\text{drift_genomic}}$ / $P_{\text{drift_pedigree}}$)

The problem here is that the reported P_{drift} values are $P_{\text{drift_pedigree}}$ when the EBVs were estimated from pedigree data and $P_{\text{drift_genomic}}$ when they were estimated from the genomic data i.e. the P_{drift} column contains both types of P_{drift} analysis. Of course we could report both $P_{\text{drift_genomic}}$ and $P_{\text{drift_pedigree}}$ for every row, by introducing an extra column, but that would make the table more complicated, and would also mean Table S7 was redundant. Instead, we have added some information to the legends of Tables S4 and S5 to explain that both types of P_{drift} value are reported. We write in the legend:

"When EBVs were estimated from pedigree data, the P_{drift} column reports the $P_{\text{drift_pedigree}}$ statistic and when genomic EBVs are estimated, the P_{drift} column reports the $P_{\text{drift_genomic}}$ statistic

(see section *Comparison of two methods to simulate evolutionary change expected under drift*, Table S7)"

Typo in Figure S1 caption: "1990-2105"

Thank you – now corrected

Reviewer(s)' Comments to Author:

Referee: 4

Comments to the Author(s).

Using genomic prediction to detect microevolutionary change of a quantitative trait, by Hunter and coworkers. The authors revisit the testing of the existence of microevolution for adult weight in the well-known Soay sheep population by comparing the evolution of observed breeding values with the ones simulated under the assumption of only drift affecting the performance for the trait. The relevant novelty is the use of genomic estimates of individual values instead of the pedigree-based EBVs. The use of genomic information for this task presents clear advantages from the past methodologies improving the accuracy of estimates and enlarging the type of populations that can be targets for such kind of studies.

We thank the reviewer for their constructive and insightful comments. We have clarified parts of the manuscript where the reviewer has indicated more context is needed. We are glad that the reviewer recognises some advantages to the approach we have taken to studying microevolutionary trends.

Notwithstanding, I have a couple of concerns on the methodology used for the study. Firstly, I think that in the particular scenario of wild populations (and many times in domestic ones) the genomic evaluation (i.e. the estimation of the GEBVs) could be better under a GBLUP framework instead of the SNP-BLUP methodology. The strategy of calculating the relationship matrix from the (mostly) genomic information and then applying classical mixed models would allow for a better use of available information. If relying exclusively on genomic data, GBLUP and SNP-BLUP methods have been proven to be equivalent. Moreover, using the single-step approach (a revision in, for example, Lourenco et al. 2002, Genes and reference therein) makes possible to include the information from all kind of individual, whatever they are genotyped or not but connected by genealogy. Genomic data allows for a more accurate estimation of actual/observed genetic relationships and a better use of within family variation (which is sometimes not possible using pedigrees). GBLUP methodology also allows for the use of more complex models like including maternal effects or repeated measures. If there is a sound sign of the trait being controlled by loci explaining different percentages of the genetic variance, this can be accounted for when constructing the GRM matrix by weighting differentially the information from each marker (see, for example, Tiezzi and Maltecca 2015, GSE).

We weren't aware of the Lourenco et al 2020 Genes paper, and thank the reviewer for bringing it to our attention. Clearly the methods described in that paper could be applicable to many populations, and the ability to include complex models with repeated measures etc is certainly

useful. However, it may not be optimal for our dataset, and certainly not for detecting microevolutionary trends, for several reasons. First, the GBLUP and SNP-BLUP methods have been shown to be equivalent, but only if it is assumed the SNP effects come from a single normal distribution. We have previously shown (Ashraf et al., 2021) that the accuracy of genomic prediction in this population is greater for models that allow the SNPs to come from a mixture of effect size distributions, with most SNPs having an effect size of 0 (e.g. the BayesB, BayesR methods). Second, the single step approaches are most advantageous when a relatively small proportion of individuals are genotyped, but the relatedness of non-genotyped individuals can be determined from pedigree data, hence boosting the sample size. Here, most of the individuals are genotyped, especially those for whom phenotype data are available. In other words, very few additional animals would be added to a model if we used a single step genomic and pedigree-derived H matrix. Finally, and this is probably the most important reason why we cannot use the single step approach described in the Lourenco et al. paper, the BLUPF90 package does not output the GEBV of each individual at each of the MCMC samples (in this case 1000 samples). This means we cannot carry forward the uncertainty of the GEBVs into the analyses that look at temporal trends, which is a major problem when trying to estimate the significance of those trends. Similarly, it does not output all of the MCMC samples of each SNP's effect size, which means we could not run the gene dropping $P_{\text{drift_genomic}}$ simulations described in the paper. In short, the advantages of using a single-step GBLUP methodology would be more than offset by the loss of some of the features we require to reliably estimate the significance of microevolutionary change. We have now cited the Lourenco paper in the manuscript, while trying to explain the approach we took. We write:

“Single-step genomic BLUP (GBLUP) methods are promising, as they can accommodate relationship matrices built jointly from SNPs and (if available) a pedigree, while also fitting non-genetic fixed and random effects (Lourenco et al., 2020). However, we are not aware of current software that does this, while also outputting the sampled MCMC iterations which are required to assess the significance of the observed microevolutionary changes. In our dataset there is little to be gained from constructing a relationship matrix that combines genotype and pedigree information as nearly all of the phenotyped animals are also genotyped.”

Secondly, I agree with the authors that in the simulation step it is better to take an approach where individual SNPs are simulated, especially when the trait is controlled by few loci of large effects. However, I wonder how robust this methodology is against departures from the real 'genetic architecture'. If the number and effects of the causal loci are poorly estimated, maybe this would lead to wrong conclusions and assuming pure infinitesimal could be a better (more conservative) option. Realise that, besides the possible major genes, most quantitative trait also present a polygenic background.

We agree that we remain uncertain about the effect that departures from the true architecture has on how well the simulations reflect patterns caused by drift. However, it is certainly not the case that assuming a pure infinitesimal model is more conservative than using an estimate of the number, genomic locations and effect sizes of segregating QTL. Intuitively, it seems reasonable to assume that cohort mean breeding values can fluctuate between generations/cohorts more when there are loci of medium-large effect segregating than when the trait is polygenic. Our data support this, because in Table S7, we show that when the simulations assume an infinitesimal model, the P values are always lower (i.e. less

conservative) than when we use a gene-dropping approach based on the estimated genetic architecture. It should perhaps be noted that our earlier exploration of the genetic architecture of weight (Ashraf et al., 2021) reported a posterior mean of ~2500 QTL, none of which were of very large effect, so our gene-dropping simulations are still treating the trait as highly polygenic. It is also worth noting that the simulations do use the posterior distribution of SNP effect sizes, so uncertainty in their effects is incorporated into the gene-dropping. We now write:

“Of course, the accuracy of the estimated SNP effects is unknown, but by using the posterior distribution of SNP effect sizes, the uncertainty in the estimates is incorporated into the drift simulations.”

I think that, at least, the influence of both items should be discussed. Further comments follow:

- Page 3, lines 48-49: This process is usually referred to as calculating the ‘genetic gain’ in the context of breeding programs.

We have avoided the term ‘genetic gain’ here because in many microevolutionary studies, the breeding values may be getting smaller (e.g. if selection favours individuals with lower values). Thus, we feel it is more inclusive to refer to testing whether breeding values have changed.

- Page 7, lines 140-141: As said before, the knowledge of the number and effect of the loci affecting a trait is not trivial.

We agree and have added a caveat. ‘acknowledging these are not easy to estimate accurately’

- Page 8, line 157: To what extent is the sampling complete?

Over 90% of lambs born in the study area are caught, tagged and sampling.

- Page 8, lines 158-159: Is mortality measured from the discovering of dead animals (frequent/extensive search in the field)? Or by the ‘absence’ of known animals in the annual control?

Both, but primarily the former. Most animals are discovered after they die (typically in one of the numerous stone, roofed structures on the island, known as Cleitan. The remaining, unaccounted for animals, are assumed dead when they are consistently absent from the ~30 censuses carried out p.a. We had already written a few lines later:

Winter mortality is monitored, with the peak of mortality occurring at the end of winter/early spring, and ca. 80% of all deceased sheep are found.

- Page 8, line 163: Even if you do not go for a in depth explanation on the way data are obtained, at least the reader should be warned/recalled that raw data are pre-corrected by some factors like sex and birth weight.

A good point. We now write:

“Weight data were pre-adjusted for non-genetic factors for the genomic prediction analyses (see below).”

- Page 8, line 169: These are realisations that weight at different ages are really different traits (i.e. with different genetic control) and, thus, can not be included in the same evaluation.

The ages where weights most obviously might have a different architecture (i.e. lambs and, to a much lesser extent, yearlings) are the ages we excluded from our analyses. By only considering adults (28 months and older), we have removed most of the effects of genetic architectures changing across ontogeny. Indeed, an earlier study showed that the genetic correlation between adult August weights at age 28 months and above was 0.99 (Wilson et al., 2007). We now point this out:

“August weights at different adult ages have genetic correlations of ~0.99 (Wilson et al., 2007).”

Of course, in animal breeding it is not uncommon for genomic prediction to be applied to datasets with animals from completely different breeds, and even then predictions are surprisingly accurate.

- Page 9, lines 180 and 182: Some units are required (percentages).

Done – 0.02 replaced with 2% and 0.95 replaced with 95%

- Page 10, lines 202-204: What about non genotyped (either with phenotype or not) but genealogically connected?

These were excluded. Without genotypes we could not estimate their GEBVs. EBVs from the pedigree data could have been estimated, but this would have resulted in an unfair comparison between GEBV and pedigree-EBV approaches.

- Page 10, line 206: Did you try to include age as a covariate? Was the reproductive status of ewes accounted for in the evaluation?

Age is included as a covariate.

“The phenotypes used in the BayesR analysis were obtained by first fitting a linear mixed model (see [26]) that included individual identity (to account for repeated measures), birth year and capture year as random effects and sex and age as fixed effects. The random effect of individual identity was used as the phenotype”

At the time of measurement (August) no ewes are pregnant. Lambs are born in April/May and conception occurs in November/December.

- Page 10, line 215: Is permanent environment effect absent from the model for genomic evaluations?

No, it was included, and estimated from the individual identity term described in the model (see previous point)

- Page 10, lines 217-219: Include some explanation on the need of different number of total and burn-in iterations in the pedigree based and genomic evaluations. Link then with the gap between sampled iterations, to end up with the same number of samples.

This is a good point. We now write:

“Note that the MCMCglmm runs required more iterations, a longer burn-in and less frequent sampling than the BayesR analyses, because more terms were estimated in the MCMCglmm models. However, both types of analysis finished with 1000 samples.”

- Page 12, line 252: Usually, beside the midparental breeding value, BVs of offspring includes a random value reflecting the Mendelian sampling, depending on the additive genetic variance for the trait and the inbreeding level of the parents. Was this term not included in the cited studies?

Thank you for bringing this point to our attention. We have checked and both the Pedantics phensim function (which we used) and the MCMCglmm rbv function (which others have used) do include a random value to reflect Mendelian sampling. We have clarified this by writing:

“Founders were assigned breeding values based on the additive genetic variance and progeny were assigned values by sampling from a Gaussian distribution with mean equal to the midparent breeding value and variance equal to half of the population additive genetic variance (for that sample of the posterior distribution of EBVs)”

- Page 12, line 261: Please, provide the rationale/argumentation for such figure.

We now cite a paper that estimates the effective population size of numerous sheep breeds, including this population (Kijas et al., 2012).

- Page 16, line 333: A space is missing before Pdrift_genomic.

Thank you. Space added.

- Page 17, line 341: But in Fig. 2e it can be seen that most of the increase is in the early years. This should be pointed out and included in the discussion of the results.

Actually, this may be a little misleading because the sample sizes were lower in the early years. The coefficients of annual change in GEBVs is remarkably consistent between the four time periods reported in Tables S4 and S5 (1980-2015, 1985-2005, 1990-2015, 2005-2015).

- Page 17, line 351: This is also observable in Figure 2b too.

Again, this does not really seem to be the case, apart from the decline in EBVs during 2005-2015, but that is likely an artefact and is explained in the supplementary section “Is the accuracy of EBV rends affected by pedigree depth”

- Page 17, lines 357-359: It is a bit awkward saying that you will present (relevant) explanations on the possible causes of the performance in the Supplementary Material. I

guess that the concept should be included in the main text and, if a formal prove exists, develop it in the Supplementary Material.

I don't think we would claim this was a formal proof of an explanation why the discrepancy between pedigree EBVs and GEBVs appears at the end of the time series. Instead, we present an analysis that strongly supports an idea raised elsewhere (Postma, 2006). The discrepancy (and its possible explanation) is not central to the paper, and will probably only interest specialists working on very similar questions. We do feel it belongs in the supplementary material, otherwise the paper would get unacceptably long. Including this section would add an extra four figures (three of which contain 39 panels) to the main paper.

- Page 18, line 365: I agree that some wild population (even some domestic ones) can not be studied with a pedigree approach because it can be poor (scarce and not reliable). However, it has to be realised that the budget for such populations uses to be limited. Any comment on the feasibility of conducting large-scale genomic studies in this kind of populations.

We suspect that constructing a pedigree in a wild population may often cost as much, or more, than a large-scale genomic study. In Soay sheep for example, it is impossible to construct a social pedigree from observing the animals because the mating system is highly promiscuous. Thus, genotyping is necessary whether using a pedigree or a genomic prediction approach (we use the same SNP chip for both parentage analysis and genomic prediction). Admittedly, it is possible to estimate social pedigrees in some nest-box breeding birds, but extra-pair paternity is very common in most species, so these social pedigrees are certain to be wrong; sometimes very wrong. One of the main motivations of this study was to show how microevolutionary changes could be detected in the absence of pedigree data; i.e. to highlight how we could finally start to expand our horizons beyond the 'usual' organisms studied by wild animal quantitative geneticists, such as nest-box breeding birds, small mammals and ungulates.

- Page 18, lines 374-375: But genomic evaluation loses accuracy the lower is the relationship between the training and the testing population.

This is true, but the training and test populations in our case were collected during the same period, on the same part of a small island population with low effective population size ($N_e \sim 200$). The reference we cite (Clark et al., 2012) shows clearly (in simulated data and a real sheep dataset) that genomic evaluation outperforms pedigree-based approaches, even when the test and training populations are ten generations apart. Here our test and training populations were alive in an almost entirely overlapping period.

- Page 19, line 392: 'Surprisingly' is, for me, a too strong word. Reword the sentence to say something like 'Contrarily to what was found in previous studies ...'.

Yes, this is a fair point. We have changed 'Surprisingly' to 'Furthermore'.

- Page 20, line 412: Noun (rate) and verb (are) do not match.

Thank you. 'Are' has been replaced with 'is'.

- Page 20, line 417: An extra comma.

Extra comma removed

- Page 20, lines 417-418: Do you think that this could be clarified/determined by extensive simulation studies?

Potentially, yes. We have added some words to this effect (see response to AE comments)

- Page 21, line 440: Said before, some consequences on the inaccurate estimation of number of loci and their effects.

We now acknowledge that the accuracy of the SNP effect sizes is unknown, but we also point out that any inaccuracy of SNP effects is incorporated into the simulations i.e. we do not treat the SNP effects as point estimates. We write:

“Of course, the accuracy of the estimated SNP effects is unknown, but by using the posterior distribution of SNP effect sizes, the uncertainty in the estimates is incorporated into the drift simulations”

- Page 21, lines 452-454: I do not totally agree. In Fig. 2e it seem that EBVs stabilise from 1995.

The temporal trends of GEBVs in the 1990-2015 and 2005-15 time periods do not support this (Tables S4, S5). The coefficients have stayed very stable at around 0.01 Kg increases per year.

- ASHRAF, B., HUNTER, D. C., BERENOS, C., ELLIS, P. A., JOHNSTON, S. E., PILKINGTON, J. G., PEMBERTON, J. M. & SLATE, J. 2021. Genomic prediction in the wild: A case study in Soay sheep. *Mol Ecol*.
- CLARK, S. A., HICKEY, J. M., DAETWYLER, H. D. & VAN DER WERF, J. H. 2012. The importance of information on relatives for the prediction of genomic breeding values and the implications for the makeup of reference data sets in livestock breeding schemes. *Genet Sel Evol*, 44, 4.
- DAETWYLER, H. D., PONG-WONG, R., VILLANUEVA, B. & WOOLLIAMS, J. A. 2010. The Impact of Genetic Architecture on Genome-Wide Evaluation Methods. *Genetics*, 185, 1021-1031.
- KIJAS, J. W., LENSTRA, J. A., HAYES, B., BOITARD, S., NETO, L. R. P., SAN CRISTOBAL, M., SERVIN, B., MCCULLOCH, R., WHAN, V., GIETZEN, K., PAIVA, S., BARENDSE, W., CIANI, E., RAADSMA, H., MCEWAN, J., DALRYMPLE, B. & INT SHEEP GENOMICS, C. 2012. Genome-Wide Analysis of the World's Sheep Breeds Reveals High Levels of Historic Mixture and Strong Recent Selection. *PLoS Biology*, 10, e1001258.
- LOURENCO, D., LEGARRA, A., TSURUTA, S., MASUDA, Y., AGUILAR, I. & MISZTAL, I. 2020. Single-Step Genomic Evaluations from Theory to Practice: Using SNP Chips and Sequence Data in BLUPF90. *Genes*, 11.
- POSTMA, E. 2006. Implications of the difference between true and predicted breeding values for the study of natural selection and micro-evolution. *Journal Of Evolutionary Biology*, 19, 309-320.
- WILSON, A. J., PEMBERTON, J. M., PILKINGTON, J. G., CLUTTON-BROCK, T. H., COLTMAN, D. W. & KRUIK, L. E. B. 2007. Quantitative genetics of growth and cryptic evolution of body size in an island population. *Evolutionary Ecology*, 21, 337-356.